

# Comparison of a Coupled Near and Far Wake Model With a Free Wake Vortex Code

Georg Pirrung[1], Vasilis Riziotis[2], Helge Madsen[1], Morten Hansen[1], and Taeseong Kim[1]

[1]Wind Energy Department, Technical University of Denmark, Frederiksborgvej 399, DK-4000 Roskilde, Denmark
[2]School of Mechanical Engineering, National Technical University of Athens, 9 Heroon Polytechneiou Str., GR15780, Athens, Greece

*Correspondence to:* Georg Pirrung (gepir@dtu.dk)

**Abstract.** This paper presents the integration of a near wake model for trailing vorticity, which is based on a prescribed wake lifting line model proposed by Beddoes, with a BEM-based far wake model and a 2D shed vorticity model. The resulting coupled aerodynamics model is validated against lifting surface computations performed using a free wake panel code. The focus of the description of the aerodynamics model is on the numerical stability, the computation speed and the accuracy of

unsteady simulations. To stabilize the near wake model, it has to be iterated to convergence, using a relaxation factor that has to be updated during the computation. Further, the effect of simplifying the exponential function approximation of the near wake model to increase the computation speed is investigated in this work. A modification of the dynamic inflow weighting factors of the far wake model is presented that ensures good induction modeling at slow time scales. Finally, the unsteady airfoil aerodynamics model is extended to provide the unsteady bound circulation for the near wake model and to improve

the modeling of the unsteady behavior of cambered airfoils. The model comparison with results from a free wake panel code and a BEM model is centered around the NREL 5 MW reference turbine. The response to pitch steps at different pitching speeds is compared. By means of prescribed vibration cases, the effect of the aerodynamic model on the predictions of the aerodynamic work is investigated. The validation shows that a BEM model can be improved by adding near wake trailed vorticity computation. For all prescribed vibration cases with high aerodynamic damping, results similar to those obtained by

the free wake model can be achieved in a small fraction of computation time with the proposed model. In the cases with low aerodynamic damping, the addition of trailed vorticity modeling shifts the results closer to those obtained by using the free wake code, but differences remain.

## 1 Introduction

This work is based on a coupled aerodynamics model, where the trailed vorticity effects in the near wake are computed based

on a model proposed by Beddoes (1987), and the far wake computation is using the well-known blade element momentum (BEM) theory. The near wake model (NWM) is a simplified prescribed wake lifting line model, which efficiently computes the induction due to the vorticity trailed during a quarter of a rotor revolution. The coupled model can be seen as a hybrid code between a traditional BEM model and the more complex vortex codes. Because a BEM model is based on an actuator disc assumption, it can not model the detailed dynamic induction response at the individual blades. Therefore the NWM



is introduced to model these unsteady induction characteristics due to load changes by pitch, eigenmotion of the blades, turbulent inflow and shear. The accuracy of the computations is improved due to the added aerodynamic coupling between airfoil sections through the trailed vorticity, alleviating the limitations of the BEM strip theory. Especially in cases with large radial load gradients, for example close to trailing edge flaps or other aerodynamic devices or close to the blade root and tip,

the cross sectional coupling will lead to an improved prediction of the steady and dynamic induction. The addition of the near wake model in an aeroservoelastic code has an acceptable effect on the total computation speed. An aeroelastic simulation with the wind turbine code HAWC2, (Larsen and Hansen, 2007; Larsen et al., 2013; Kim et al., 2013), of the DTU 10 MW turbine (Bak et al., 2012) in normal operation with turbulent inflow takes roughly 10% (30 aerodynamic sections) to 40% (55 aerodynamic sections) longer if the near wake model is enabled than if a pure BEM model is used.

The coupled model using the modified BEM approach for the far wake has been proposed by Madsen and Rasmussen (2004) and extended by Andersen (2010). Further improvement has been presented by Pirrung et al. (2012), where an iterative procedure was used to ensure convergence and avoid numerical instabilities of the NWM. An application of the coupled model to estimate the critical flutter speeds of the NREL 5MW turbine (Jonkman et al., 2009) also including blades with modified stiffness, has been described by Pirrung et al. (2014), where the coupled aerodynamics model has predicted 4-10 % higher

critical flutter speeds than the unsteady BEM model in the aeroservoelastic wind turbine code HAWC2.

In the present paper, the iteration procedure of the NWM used by Pirrung et al. (2012) is presented in more detail, as well as a method to compute the necessary relaxation factor during a simulation, removing the need for additional input or very conservative relaxation factors that are independent of spatial and temporal discretization and increase the computation time. Further, the NWM is simplified to accelerate the computations with small loss of accuracy of the unsteady results.

The dynamic responses to pitch steps and prescribed blade vibrations are validated by comparing them to results from the more complex free wake code GENUVP (Voutsinas, 2006). The focus in the pitch step cases is the dynamic induction response, while the prescribed vibration cases are evaluated based on aerodynamic work during a period of oscillation. It is found that the coupled aerodynamic model is capable of producing results that agree much better with results obtained from the free wake code than the unsteady BEM model in most cases, without a dramatic increase in computation time. The more

accurate computation of aerodynamic work can have a considerable impact on the aeroelastic response in the case where the total damping is close to zero, such as for edgewise vibrations.

This paper is structured as follows: In the next section a short description of the NWM and a previous implementation of the coupling to a far wake model and shed vorticity model are presented. In Sect. 4, modifications to far wake and shed vorticity model are proposed to improve the interaction of these models with the near wake model and to increase the accuracy of the

dynamic lift computation for cambered airfoils. This is followed by a description of the iterative procedure to stabilize the near wake model in Sect. 5. A way of simplifying the NWM to accelerate the computation is presented in Sect. 6. In Sect. 7 the free wake panel code used for validation of the coupled near and far wake model is briefly described. The effects of the model modifications and results from the code comparison are shown and discussed in Sect. 8.



## 2 Original Model description

The structure of the previous implementation (Madsen and Rasmussen, 2004; Andersen, 2010) of the model is shown in Fig. 1. From the velocity triangle, denoted as $VT$, follows a geometric angle of attack (AOA) $\alpha_{QS}$ and a relative velocity $v_r$. An effective AOA $\alpha_{eff}$ is obtained through a 2D modeling of the shed vorticity effects, which is briefly described in Sect. 2.3.

This effective AOA is used to determine the aerodynamic forces and the thrust coefficient $C_T$. The thrust coefficient leads to a far wake induction factor $a_{FW}$, requiring a coupling factor $k_{FW}$ as input. Section 2.2 contains the dynamic inflow model, using the weighting factors $A_1$ and $A_2$, which is used to determine the unsteady far wake induction $u_{FW,dyn}$.

   Using this far wake induction, and the near wake induction from the previous time step, a new intermediate velocity triangle $VT_i$ is determined, with a new quasi steady AOA and relative velocity. These lead to the bound circulation $\Gamma_{QS}$. The difference

in $\Gamma_{QS}$ between adjacent blade sections, denoted as $\Delta\Gamma$ in the following, determines the trailed vorticity. In the next section it is shown how the induced velocity $W$ due to the near wake, which is added to $u_{FW}$ to obtain the total induced velocity $u_{tot}$ at each blade section, follows from the trailed vortices. The total induced velocity will then, in addition to the relative velocity due to blade motion and turbulence in the incoming wind, determine the velocity triangle after the time step $\Delta t$.

### 2.1 Near wake model

The NWM enables a fast computation of the induction due to the trailed vorticity behind a rotor blade. The trailed wake can be discretized into trailed vortex arcs from several positions on the blade, where each arc consists of a number of vortex elements. The induction at a blade section due to each vortex element can be computed using the Biot-Savart law, but this computation is numerically expensive as the influence of each vortex element on the induction at each blade section has to be determined. Beddoes (1987) proposed to avoid these expensive computations by assuming that the trailed vorticity follows circular vortex

arcs in the rotor plane and limiting the computation to a quarter rotation. In this quarter rotation, the axial induction $dw$ from a vortex element at a blade position is decreasing as the vortex element moves away from the blade, starting with a value $dw_0$. This decreasing induction, following from the Biot-Savart law, is approximated by exponential functions:

$$\frac{dw}{dw_0} \approx 1.359e^{-\beta/\Phi} - 0.359e^{-4\beta/\Phi}, \tag{1}$$

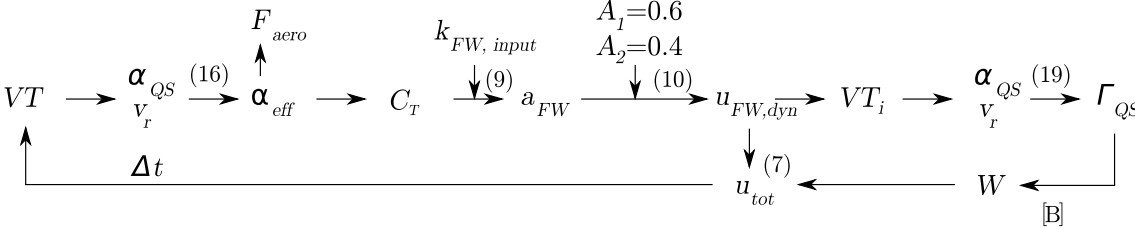

**Figure 1.** The previous implementation of the coupled near and far wake model, as described by Madsen and Rasmussen (2004) and Andersen (2010). The numbers in parenthesis refer to the equations in the following sections, [B] to the original model by Beddoes (1987).



where $\Phi$ is a geometric factor depending on the radius from which the vortex is trailed and the distance between the vortex trailing point and the blade section where the induction is computed. The angle $\beta$ determines how much the blade has rotated away from the vortex element. The numerically efficient trailing wake algorithm gives the induction $W$ due to the trailed vorticity at time step $i$ at a blade section $s$ as:

$$W_s^i = \sum_{v=1}^{N_v} W_{s,v}^i, \tag{2}$$

where $N_v$ is the number of vortex arcs trailed from the blade and $W_{s,v}$ is the induction due to a single vortex arc $v$ at the blade section. It consists of components $X_{s,v}^i$ and $Y_{s,v}^i$ corresponding to both of the exponential terms in Eq. (1):

$$W_{s,v}^i = X_{s,v}^i + Y_{s,v}^i, \tag{3}$$

$$X_{s,v}^i = X_{s,v}^{i-1} e^{-\Delta\beta_v/\Phi_{s,v}} + D_{X,s,v}\Delta\Gamma_v(1 - e^{-\Delta\beta_v/\Phi_{s,v}}), \tag{3a}$$

$$Y_{s,v}^i = Y_{s,v}^{i-1} e^{-4\Delta\beta_v/\Phi_{s,v}} + D_{Y,s,v}\Delta\Gamma_v(1 - e^{-4\Delta\beta_v/\Phi_{s,v}}), \tag{3b}$$

where $\Delta\Gamma_v$ is the trailed vortex strength, which depends on the radial difference in bound circulation between the blade sections adjacent to the vortex trailing point. The relative movement of the blade in the rotor plane during the time step at the vortex trailing point is denoted as $\Delta\beta_v = (v_{r,in-plane}/r)\Delta t$. The in-plane velocity component perpendicular to the lifting line is denoted as $v_{r,in-plane}$. Equations (3a) and (3b) show that the induction consists of a decreasing part of the induction at the previous time step, due to the previously trailed wake moving away from the blade, and the contributions from the newest element, given by the $D_{X,s,v}$ and $D_{Y,s,v}$ terms. These equations are less time step sensitive and computationally faster than the original equations by Beddoes. They have been proposed by Pirrung et al. (2016), as well as a correction for the helix angle of the trailed vorticity. Pirrung et al. (2016) also describe how the tangential induction is computed based on the same approach.

### 2.2 Coupling to far wake model

The NWM, which only computes a fraction of the total rotor induction, is complemented by a modified BEM model for the far wake. The total induced velocity at a blade section is computed as

$$u_{tot} = u_{FW} + W, \tag{4}$$

where $u_{FW}$ is the far wake component of the induced velocity and $W$ is the near wake component, cf. Eq. (2).

The far wake component $u_{FW}$ is computed based on the BEM model implementation in HAWC2 that uses a polynomial to relate the local thrust coefficient with the axial induction factor at each annular element. To account for the near wake induction, the far wake induction factor $a_{FW}$ is computed without a tip loss correction and based on a thrust coefficient $C_T$ that is reduced by the coupling factor $k_{FW}$. The coupling factor is automatically adjusted during the computation to closely match the thrust of a reference BEM model, where the induction factor $a_{ref}$ is computed including tip loss effects (Pirrung et al., 2016).



To account for the far wake dynamics, this work uses the dynamic inflow model implemented in HAWC2. Two parallel first order low pass filters are applied on the quasi steady induced velocities $u_{FW,QS} = a_{FW}u_\infty$ from the BEM model:

$$u^i_{FW,dyn} = A_1 u^i_1 + A_2 u^i_2 \tag{5}$$
$$u^i_1 = u^{i-1}_1 e^{-\Delta t/\tau_1} + u^i_{FW,QS}(1 - e^{-\Delta t/\tau_1}) \tag{6}$$
$$u^i_2 = u^{i-1}_2 e^{-\Delta t/\tau_2} + u^i_{FW,QS}(1 - e^{-\Delta t/\tau_2}). \tag{7}$$

In a pure BEM computation and the previous far wake model implementation, the factors $A_i$ are $A_1 = 0.6$ and $A_2 = 0.4$. They are used to divide the induction into a faster and slower reacting part, corresponding to a faster time constant $\tau_1$ and the slower time constant $\tau_2$. Both time constants are a function of radius and mean loading. The constants $A_i$ and $\tau_i$ have been tuned to the actuator disc simulations of step changes in uniform loading (Sørensen and Madsen, 2006).

## 2.3 Unsteady airfoil aerodynamics model

The sketch in Fig. 2 illustrates how the shed vorticity due to the time variation of the bound circulation induces a downwash $w_{3/4}$ at the three quarter chord of an airfoil. This downwash will change the angle of attack and thus the lift, drag and moment coefficients according to the airfoil polars, as well as the directions of the aerodynamic forces. The inviscid part of the unsteady airfoil aerodynamics model by Hansen et al. (2004) treats the shed vorticity effects as a time lag on the angle of attack according to Jones' function for a flat plate. The effective angle of attack $\alpha_{eff}$, which determines the magnitude and direction of the unsteady aerodynamic forces, is computed as:

$$T^i_0 = \frac{c}{2v^i_r} \tag{8}$$
$$x^i_1 = x^{i-1}_1 e^{-0.0455\frac{\Delta t}{T^i_0}} + \frac{1}{2}(\alpha^i_{QS} + \alpha^{i-1}_{QS})0.165 v^i_r(1 - e^{-0.0455\frac{\Delta t}{T^i_0}}) \tag{9}$$
$$x^i_2 = x^{i-1}_2 e^{-0.3\frac{\Delta t}{T^i_0}} + \frac{1}{2}(\alpha^i_{QS} + \alpha^{i-1}_{QS})0.335 v^i_r(1 - e^{-0.3\frac{\Delta t}{T^i_0}}) \tag{10}$$
$$\alpha^i_{eff} = \frac{1}{2}\alpha^i_{QS} + (x^i_1 + x^i_2)/v^i_r, \tag{11}$$

where the superscript $i$ denotes the time step and $c$ the chord length. Further, $\alpha_{QS}$ is the quasi steady angle of attack resulting from the velocity triangle at the blade section and $v_r$ denotes the corresponding relative velocity.

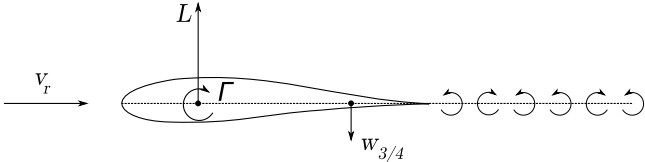

**Figure 2.** Cambered airfoil in parallel inflow to the chord line. The shed wake corresponds to the time history of the bound circulation.



## 3   Model overview

The structure of the current implementation of the coupled near and far wake model is shown in Fig. 3. The changes to the previous implementation, cf. Fig. 1 are:

- The weighting factors $A_i$ of the far wake dynamic inflow are adjusted during the computation to account for the induction computed by the near wake model, which is explained in Sect. 4.1.

- The trailed vorticity is no longer based on the quasi steady bound circulation $\Gamma_{QS}$, but instead on a dynamic bound circulation $\Gamma_{dyn}$. The computation of the dynamic bound circulation is shown in Sect. 4.2.1.

- The near wake induction is computed in an iteration loop, which is detailed in Sect. 5.

- The coupling factor is no longer needed as input, but instead continually updated during the computation, as described by Pirrung et al. (2016).

- The trailed vorticity is assumed to follow helix arcs to account for the downwind convection of the trailed vorticity. To achieve this, $\Phi$ is multiplied with a correction function $f$, depending on the blade section and vortex trailing point, as well as the helix angle at which the vortex is trailed (Pirrung et al., 2016).

- The computation of $\alpha_{eff}$ according to shed vorticity effects is improved for cambered airfoils, which is explained in Sect. 4.2.2.

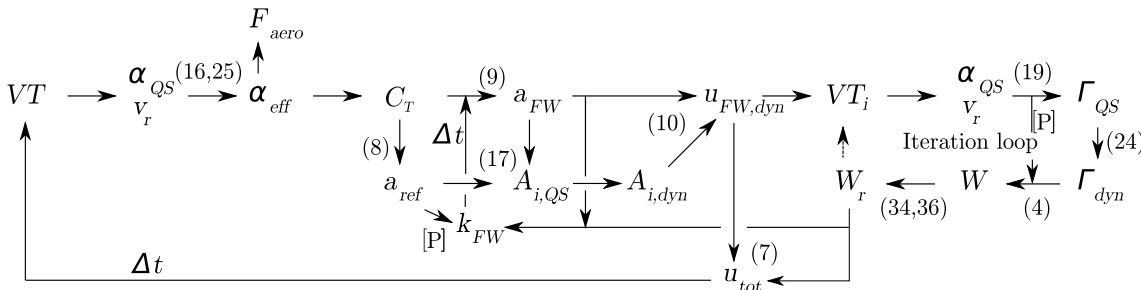

**Figure 3.** Overview of one time step in the coupled near and far wake model used in this work. Relevant equation numbers are included. [P] refers to Pirrung et al. (2016).



## 4 Modifications to far wake and shed vorticity model

### 4.1 Adapting the weighting of the dynamic inflow time filters

The dynamic inflow model described in Sect. 2.2, which has been tuned for BEM computations, has to be modified if a part of the induction is covered by the NWM. The objective is to obtain a similar slow induction response with the coupled near and far wake model as with an unsteady BEM model.

This requires a modification of the constants $A_1$ and $A_2$ in Eq. (5). The new constants $A_i$ are computed based on the far wake induction factor $a_{FW}$ and a reference induction factor obtained from a BEM model with tip loss correction, cf Sect. 2.2. The weighting constants for the far wake model are determined such that roughly 40 % of the total induction are considered to be reacting slowly, as in the original dynamic inflow model for BEM computation, Eq. (5):

$$A_{1,FW} = \frac{0.4 a_{ref}}{a_{FW}} \tag{12}$$

$$A_{2,FW} = 1 - A_{1,FW}. \tag{13}$$

The factors are continuously updated during the computations. A first order low pass filter with the far wake time constant $\tau_2$ of the dynamic inflow model is applied on $A_{1,FW}$ to make sure this model does not introduce unphysical rapid induction variations due to instantaneous changes of the weighting factors.

### 4.2 Extensions of the unsteady airfoil aerodynamics model

#### 4.2.1 Unsteady circulation computation

The influence of shed vorticity on the bound circulation buildup has to be be considered when determining the strength of the trailed vortices of the NWM. Joukowski's relation between quasi steady lift $L_{QS}$ and circulation $\Gamma_{QS}$,

$$\Gamma_{QS} = \frac{L_{QS}}{\rho v_r} = \frac{1}{2} v_r c C_L, \tag{14}$$

which has been used by Madsen and Rasmussen (2004) and Andersen (2010) to determine the bound vorticity, is not valid for unsteady conditions. The error of calculating the circulation based on the unsteady lift at an airfoil section depends on the reduced frequency $k = \omega c/(2v_r)$, where $\omega$ is the angular velocity, $c$ is the chord length, and $v_r$ is the relative flow speed. For an airfoil pitching harmonically about the three-quarter chord point, the error has been estimated by Madsen and Gaunaa (2004) to be 10% at $k = 0.1$ and 100 % at $k = 0.8$, which for the NREL 5 MW reference turbine at rated wind and rotor speed corresponds to frequencies of about 1.2 and 9.8 Hz at 60 m rotor radius with a chord of 2 m. Except for the first flapwise and edgewise bending frequencies, most relevant modal frequencies for modern blades are between these values, which shows that it is important to include a modeling of the unsteady circulation.





In this paper, the step response of the circulation is approximated by the three term indicial function used by Madsen and Gaunaa (2004).

$$\Gamma_{dyn}/\Gamma_{QS} = 1 - A_{\Gamma,1}e^{-b_{\Gamma,1}\tau} - A_{\Gamma,2}e^{-b_{\Gamma,2}\tau} - A_{\Gamma,3}e^{-b_{\Gamma,3}\tau}, \quad \text{where} \tag{15}$$

$$\tau = \Delta t \frac{2v_r}{c}, \quad A_{\Gamma,1} = 0.5547, \quad A_{\Gamma,2} = 0.1828, \quad A_{\Gamma,3} = 0.2656, \tag{16}$$

$$b_{\Gamma,1} = 0.3064, \quad b_{\Gamma,2} = 0.0439, \quad b_{\Gamma,3} = 3.227 \tag{17}$$

The algorithm is implemented analogue to the computation for the effective angle of attack in Equations (8)-(11):

$$x_{\Gamma,j}^i = x_{\Gamma,j}^{i-1}e^{-b_{\Gamma,j}\frac{\Delta t}{T_0^i}} + \frac{A_{\Gamma,j}}{2}(\Gamma_{QS}^i + \Gamma_{QS}^{i-1})(1 - e^{-b_{\Gamma,j}\frac{\Delta t}{T_0^i}}) \tag{18}$$

$$\Gamma_{dyn}^i = x_{\Gamma,1}^i + x_{\Gamma,2}^i + x_{\Gamma,3}^i, \tag{19}$$

where the quasi steady circulation is computed from the quasi steady lift coefficient using Eq. (14).

### 4.2.2 Unsteady aerodynamics of cambered airfoils

Any change in bound circulation $\Gamma$, which is a function of $v_r C_L$, cf. Eq. (14) should lead to the corresponding shed vorticity. The implementation of the shed vorticity model according to Hansen et al. (2004), cf. Equations (9-11) is based on the term $\alpha_{QS}v_r$. The camber of the airfoil is neglected in this computation of the shed vorticity effects. We propose in this work to replace $\alpha_{QS}$ in Equations (9 to 11) by $\alpha_{QS,camber}$, with

$$\alpha_{QS,camber} = \alpha_{QS} - \alpha_0, \tag{20}$$

where $\alpha_0$ is the zero lift angle of the airfoil.

The impact of this modification is shown for basic cases of relative velocity changes in Figures 4 and 5, where an airfoil with a $2\pi$ lift gradient, a 2 m chord length, a zero lift angle of $-3°$ and a drag coefficient of $C_D = 0.005$ has been simulated. The airfoil characteristics and chord length have been chosen to be similar to the outboard region of the NREL 5MW reference turbine and the geometric angle of attack has been chosen as zero, to show the isolated effect of airfoil camber. In Fig. 4, the variation of effective angle of attack due to a step change of relative speed from 70 m/s to 71 m/s within a time step of 0.01 s is shown. Without the effect of camber , the change in relative speed has no influence on the angle of attack, because $\alpha_{QS}$ is a constant zero. The effect of camber leads to a lower angle of attack due to the shed vorticity caused by the increase in bound circulation. The camber effect is small, and the angle of attack changes only by less than 0.02 degrees immediately after the relative speed step. In Fig. 5 the induced drag due to angle of attack changes is compared to the viscous drag in case of a vibration of the airfoil section parallel to the inflow. There would be no induced drag in this example if camber was excluded from the effective angle of attack computation. The amplitude of the vibration is 1 m, the frequency 1 Hz. The effect of induced drag is of the same order of magnitude as the airfoil drag, which indicates the importance of including the airfoil camber in the unsteady airfoil aerodynamics model. The camber effect is included in all further computations presented in this paper except in the left plot of Fig. 18, where it is excluded to investigate its importance on in-plane blade vibrations.




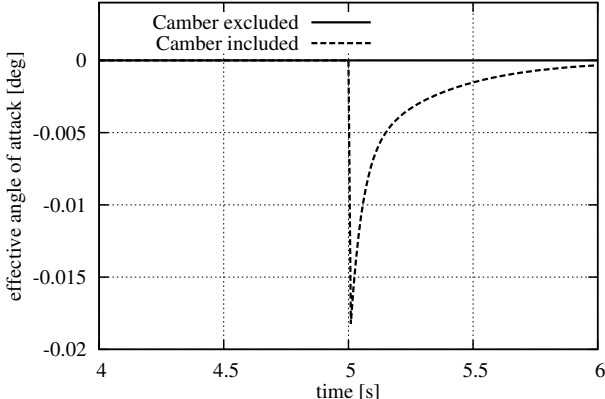
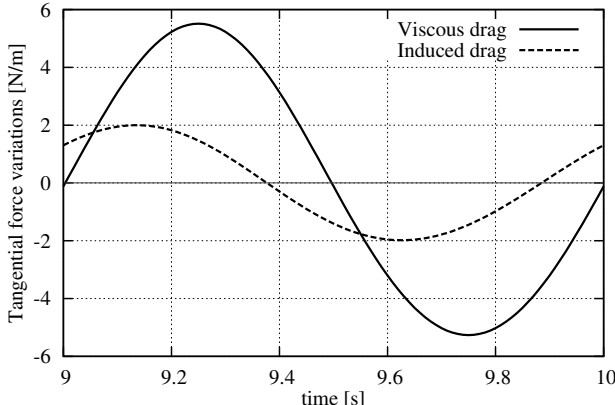

**Figure 4.** Effect of including camber in the unsteady aerodynamics model on effective angle of attack during a step in relative velocity.

**Figure 5.** Comparison of viscous drag and induced drag during oscillations of a cambered airfoil parallel to the inflow at 1 Hz. The effect of camber is included as proposed in Eq. (20). The mean drag has been subtracted.

In the unsteady circulation computation described in the previous section, the camber is accounted for through the quasi steady circulation $\Gamma_{QS}$, which is based on the lift coefficient, cf. Eq. (14).

## 5   Iterative near wake and shed vorticity model

### 5.1   Iteration scheme

The NWM can become numerically unstable depending on the time step, operating point of the turbine, blade geometry and radial calculation point distribution (Pirrung et al., 2012). Fig. 7 shows the maximum time step where a stable computation is possible for a fine and coarse geometry definition, shown in Fig. 6, of the NREL 5 MW blade. The coarse geometry definition is a blade geometry typically distributed for BEM computations and the fine distribution is more suitable for computations with higher fidelity codes. The aerodynamic calculation points and vortex trailing points follow a cosine distribution, which means

they are placed at equi-angle increments. The time steps have been determined in a numerical experiment, where the time step has been decreased until large oscillations of the induction disappear. The results are accurate to the first significant digit. It can be seen that the finer blade geometry leads to a more stable computation. This can be explained by the smoother blade tip, where the blade chord is approaching zero. Thus the radial circulation gradient at the very blade tip is smaller and the vortex strength of the tip vortex is distributed to several weaker trailed vortices in the tip region that are less likely to cause numerical

instabilities. In a coupled aeroelastic simulation, the small stable time steps for resolutions of 30 to 60 points would lead to a very slow computation especially in case of the coarser blade geometry.

The numerical instability which occurs at larger time steps can be explained as follows: The axial induction due to trailed vortices typically reduces the angle of attack at a blade section, which in attached flow leads to a reduced lift. In the original


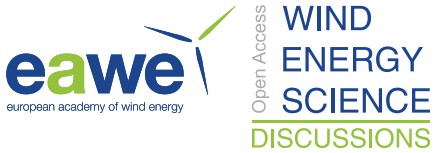

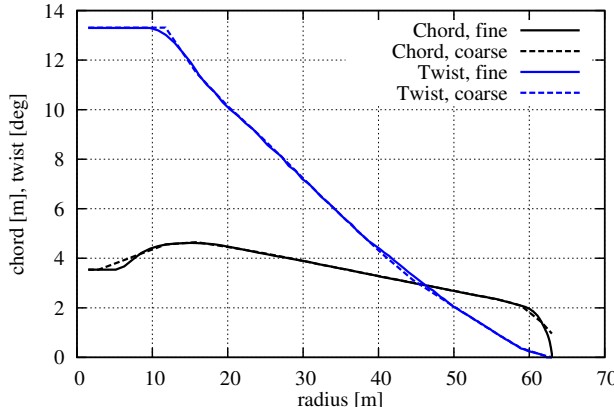
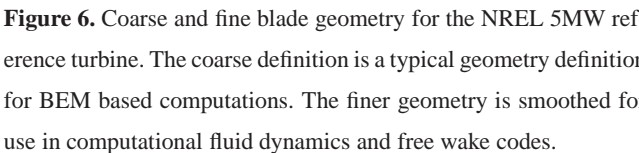

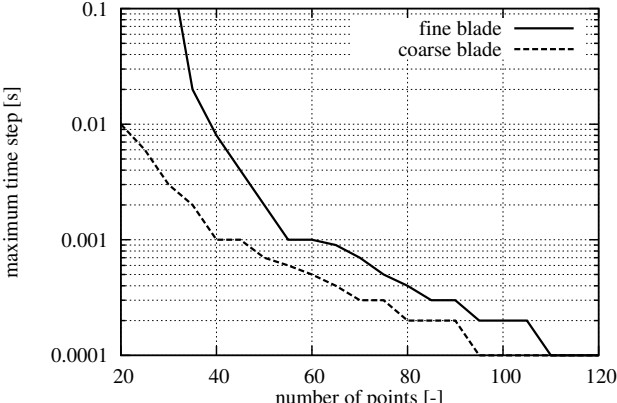

**Figure 6.** Coarse and fine blade geometry for the NREL 5MW reference turbine. The coarse definition is a typical geometry definition for BEM based computations. The finer geometry is smoothed for use in computational fluid dynamics and free wake codes.

**Figure 7.** Maximum stable time step depending on number of points for the coarse and fine blade geometries of the NREL 5 MW reference turbine. The points are distributed using a full cosine distribution (Pirrung et al., 2012). The results are obtained through numerical experiment.

implementation of the NWM the constant circulation trailed during a time step is only depending on the flow conditions at the blade at the beginning of a time step. Thus a longer time step will lead to a bigger induction and thus a further reduction in lift in the next time step. If the time step is too large, the induction can become big enough to create a negative lift in the next time step, that is bigger in absolute value than the previous positive lift. This in turn leads to stronger trailed vortices of opposite

sign, which will cause even bigger induced velocities in the opposite direction, which again leads to stronger vortices.

To stabilize the NWM the balance between trailed vortex strength based on the sectional circulation and the induced velocities are iterated to equilibrium in each time step, which removes the need for small time steps to stabilize the aerodynamics model. The iteration is structured as follows:

**1** The quasi-steady circulation is computed according to Joukowski's law using the velocity triangle at the airfoil section based
on the induction from the last iteration.

**2** The unsteady circulation is computed including shed vorticity effects, cf. Sect. 4.2

**3** This unsteady circulation defines the constant vortex strengths trailed during a time step

**4** These constant vortex strengths lead to an induction at all airfoil sections.

**5** The new induction is combined from the inductions from step 1 and 4 by applying a relaxation factor: $W_i = W_{i-1}f_r + $
$W_i(1 - f_r)$, where the subscript $i$ indicates the iteration number. If $W_i$ is sufficiently close to $W_{i-1}$, it is the desired converged induction.




The BEM model for the far wake is excluded from this iteration procedure. The AOA and relative velocity used to compute the far wake induction are the values from the converged iteration in the previous time step. This is accelerating the computation and is feasible because the near wake effects are on a much faster time scale than the dynamic inflow effects in the BEM model.

**5.2   Estimation of the necessary relaxation factor**

5   In the following, an estimation of the relaxation factor for a blade section is described. A conservative estimation is based on the least stable case which is characterized by the following properties:

– One single blade section with one vortex trailing from each side. Adjacent sections would tend to have similar circulations and therefore reduce the vortex strengths and the corresponding induction at the blade section. The trailed vortices on both sides of the section depend only on the bound circulation $\Gamma$ of that section.

10   – The lift coefficient is linearly dependent on the angle of attack, $C_L = 2\pi\alpha$. A reduced but still positive gradient due to stall would stabilize the model.

– No prior trailed vorticity is present. It would stabilize the model, because the induction would not only be determined by the momentary circulation at the section, but also by the decaying influence of the wake trailed before. If the model converges in the very first time step, with a given induction at the section from the previous iteration then the iterations 15   will also converge with prior trailed vorticity.

– The helix angle at which the vortices are trailed is assumed to be small. Thus all the induction due to trailed vorticity is assumed to be axial induction.

With these assumptions, the downwash after a time step $\Delta t$ can be determined by summing up the contributions of the newest element, cf. the right terms in Equations (3a) and (3b), for both adjacent vortices:

$$W_i = \sum_{v=1}^{2}(-1)^v\Gamma(D_{X,v}(1 - e^{-\Delta\beta/\Phi_v}) + D_{Y,v}(1 - e^{-4\Delta\beta/\Phi_v})), \tag{21}$$

where the subscript $v$ denotes the vortex further inboard ($v = 1$) and outboard ($v = 2$) of the section with the bound circulation $\Gamma$. The subscript $i$ denotes the iteration. Because the tangential induction is neglected, $\Delta\beta$ is only a function of the rotation speed of the turbine and the time step. Thus $\Gamma$ is the only variable in Eq. (21) that depends on the induction at the section:

$$\Gamma = \frac{\Gamma_{dyn}}{\Gamma_{QS}}\frac{1}{2}cC_Lv_r \tag{22}$$

$$= \frac{\Gamma_{dyn}}{\Gamma_{QS}}c\pi v_r\alpha \tag{23}$$

$$= \frac{\Gamma_{dyn}}{\Gamma_{QS}}c\pi\sqrt{(v_\infty - W_{i-1})^2 + (\Omega r)^2}\arctan\left(\frac{v_\infty - W_{i-1}}{\Omega r}\right), \tag{24}$$

where $v_\infty$ is the free wind speed and the step response function from Eq. (15) evaluated at half a time step gives $\Gamma_{dyn}/\Gamma_{QS}$ because we consider the first time step, thus a buildup of the circulation from zero.



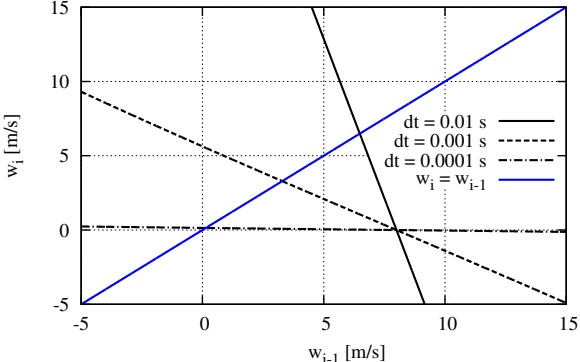

**Figure 8.** Downwash after an iteration as a function of the downwash from the preceding iteration in case of a single section with trailed vortices.

Equation (21) is plotted for different time steps as a function of $W_{i-1}$, the induction from the previous iteration, for the blade tip section of the NREL 5MW reference turbine at 8 m/s wind speed and 9.2 rpm rotor speed, corresponding to a tip speed ratio of 7.6, in Fig. 8. The airfoil camber is neglected.

The intersections of the curves with the blue curve ($W_i = W_{i-1}$) are the converged solutions, where a new iteration would

lead to exactly the same induction as the previous one. The calculations for different time steps have different converged inductions, because the length of the trailed vortex filaments is proportional to the time step. But not only the converged solution changes, also the gradient of the curves, which leads to a condition for convergence: If the distance from the converged solution decreases during a time step,

$$|W_i - W_{conv}| < |W_{i-1} - W_{conv}|, \tag{25}$$

the iterative process converges. As seen in Fig. 8, the gradient of the curves is almost independent of $W_{i-1}$. The gradients are negative because induction reduces the angle of attack. Therefore an approximation of condition (25) can be used:

$$\frac{dW_i}{dW_{i-1}} > -1. \tag{26}$$

This gradient can be derived from Equations (21) and (24) as:

$$\frac{dW_i}{dW_{i-1}} = \frac{\Gamma_{dyn}}{\Gamma_{QS}} \pi c (B_1 - B_2) \left( \frac{\alpha(v_\infty - W_{i-1})}{v_r} + \frac{v_r}{\Omega r \left( \left( \frac{v_\infty - W_{i-1}}{\Omega r} \right)^2 + 1 \right)} \right), \quad \text{where} \tag{27}$$

$$B_v = (-1)^v (D_{X,v}(1 - e^{-\Delta\beta/\Phi_v}) + D_{Y,v}(1 - e^{-4\Delta\beta/\Phi_v})) \tag{28}$$

The gradient is mainly depending on the time step and point density (through $B_1$ and $B_2$) and the rotational speed.





Instead of reducing time step and point density until a simulation is stable, which can lead to time steps orders of magnitude smaller than commonly used in aeroelastic codes and low spatial resolution, a relaxation factor $f_r$ can be introduced, so that:

$$W_{i,r} = W_i(1 - f_r) + W_{i-1}f_r. \tag{29}$$

The derivative of this downwash with regard to the old downwash is:

$$\frac{dW_{i,r}}{dW_{i-1}} = \frac{dW_i}{dW_{i-1}}(1 - f_r) + f_r. \tag{30}$$

For the minimum relaxation factor $r$, that allows for a stable computation ($dW_{i,r}/dW_{i-1} = -1$), follows:

$$f_r = -\frac{1 + \frac{dW_i}{dW_{i-1}}}{1 - \frac{dW_i}{dW_{i-1}}}, \tag{31}$$

which can be determined depending on the time step $\Delta t$, the point distribution, and the number of points on the blade.

In the initial phase of the simulation, the maximum relaxation factor for all blade sections can be quickly determined by setting $W_{i-1} = 0$ in Eq. (27) and looping through the sections. The highest necessary relaxation factor for one section that has been found is then used for the whole blade. As the simulation continues, the relaxation factor can be updated whenever there are big changes in rotational speed, induction, or blade pitch. If the relaxation factor is updated every several time steps, determining the relaxation factor takes negligible computation time. Choosing a slightly more conservative relaxation factor than what has been estimated will ensure stability also in different conditions than the ones the factor was based on.

## 6   Accelerating the NWM

In this section, an approach to accelerate the model is presented. The number of exponential terms used to approximate the decreasing induction with increasing distance from the blade in Eq. (1) is reduced to one. Using only one exponential term removes the $Y_w$ component in the near wake algorithm, Eq. (3b) and thus halves the computation time.

The reduced approximation function is defined as:

$$\frac{dw}{dw_0} \approx 1.359e^{-\beta/\Phi} - 0.359e^{-4\beta/\Phi} \approx A^*e^{-\beta/\Phi^*}. \tag{32}$$

The values of $A^*$ and $\Phi^*$ are found by solving the following equations:

$$W(\beta = \infty) = \int_0^\infty 1.359e^{-\beta/\Phi} - 0.359e^{-4\beta/\Phi}d\beta = \int_0^\infty A^*e^{-\beta/\Phi^*}d\beta \tag{33}$$

$$\int_0^\infty W(\beta = \infty) - W(\beta)d\beta = \int_0^\infty \Phi(1.359e^{-\beta/\Phi} - \frac{0.359}{4}e^{-4\beta/\Phi})d\beta = \int_0^\infty \Phi^*(A^*e^{-\beta/\Phi^*})d\beta. \tag{34}$$

Equation (33) ensures that the quasi steady induction $W(\beta = \infty)$ of the reduced model is equal to the one computed by the original model for a trailed vortex with constant strength. Equation (34) ensures a good dynamic behavior by requiring the



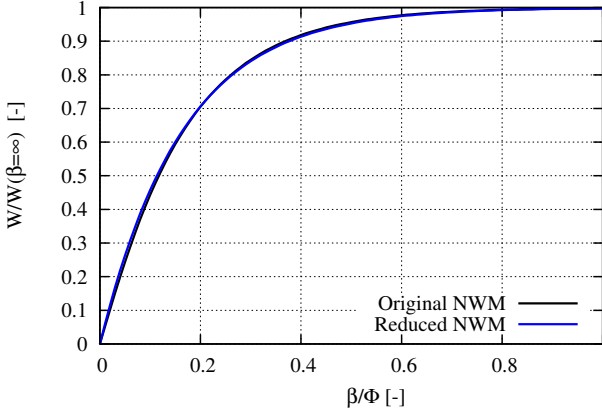

**Figure 9.** Comparison of induction buildup between full NWM and reduced NWM, depending on the length of a trailed vortex filament with constant circulation.

time integral of the difference between dynamic and quasi steady induction to be identical to the original model. The solution to these equations is

$$A^* = \frac{(1.359 - 0.359/4)^2}{1.359 - 0.359/16}, \qquad \Phi^* = \Phi \frac{1.359 - 0.359/16}{1.359 - 0.359/4} \tag{35}$$

A comparison of the buildup of induction in time, corresponding to the integral of the exponential functions, is shown in Fig. 9. The largest deviations of the reduced model from the original model are below 2.5 % of the quasi steady induction $W(\beta = \infty)$.

## 7   Free wake code

GENUVP is a potential flow solver combining a panel representation of the solid boundaries (blades) with a vortex particle representation of the wake. In the present work, the blades are considered as thin-lifting surfaces carrying piecewise constant dipole distribution (equivalent to horseshoe type vortex filaments). Blades shed vorticity in the wake along their trailing edges and their tips (vorticity emission line). In the model a hybrid wake approach is followed. The near wake part, consisting of the newly shed vorticity trailed within the current time step, is modelled as a vortex sheet also carrying piecewise constant dipole distribution. Within every time step, a strip of wake panels is released that are in contact with the emission line. Applying the no-penetration boundary condition at the centre of each solid panel and the Kutta condition along the emission line the unknown dipole intensities are determined. Then at the end of each time step, the newly shed vorticity is transformed into vortex particles and then all vortex particles are convected downstream with the free flow velocity (free wake representation) into their new positions. The layout of the modelling is shown in Fig. 10. Details of the model can be found in Voutsinas (2006).

Since GENUVP is defined as a potential flow solver, the loads need correction in order to account for viscous effects. This is done by means of the generalized ONERA unsteady aerodynamics and dynamic stall model (Petot, 1989). The potential load




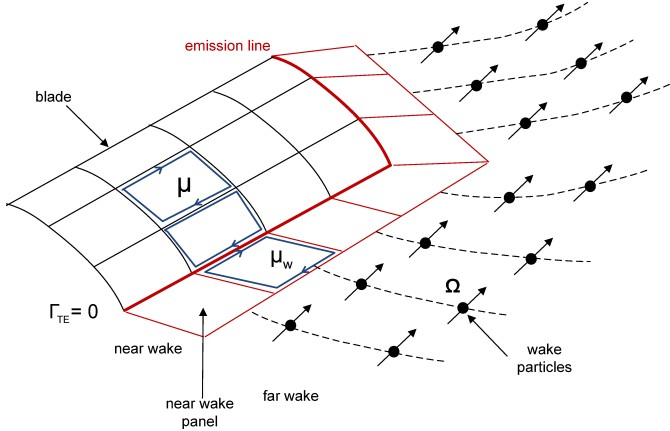

**Figure 10.** Layout of the free-wake modelling of a blade: black lines
define the blade surface panels; red lines define the wake generated
within a time step; symbols represent freely moving particles.

is calculated by integrating pressures (pressure differences between pressure and suction side) over the lifting surfaces. Then, through a consistent definition of the local flow angle of attack and relative flow velocity corrections are applied on the potential loads on the basis of the ONERA model. It is noted that the ONERA model splits the aerodynamic forces into a potential and a separated flow component. This is done through the introduction of two equivalent circulation parameters defined both for the

5 lift and the drag force. In GENUVP and in case of attached flow conditions no correction is applied on the unsteady lift force computed by the free wake code. The only correction applied is the inclusion of the viscous drag contribution to the loads. In case of separated flow conditions the separated flow component of the ONERA model is superimposed to the potential loads provided by the free wake model (Riziotis and Voutsinas, 1997).

In the case of aeroelastic coupling, the aerodynamic part will receive the deformed geometry and the deformation velocity

and feedback the loading. The deformed geometry as well as the deformation velocities are introduced into the boundary conditions and therefore the flow is accordingly adjusted.

The GENUVP free wake code has been thoroughly validated over the past years against measured data both on wind turbines and helicopter rotors in the framework of numerous EU funded projects. Blade loads and wake velocities comparisons against measurements have been performed on the MEXICO rotor in the context of Innwind.eu project (Madsen et al., 2015).

Moreover, detailed blade load calculations have been performed for the NREL test rotor and results have been compared to experimental data (NREL experiment) and CFD computations (Chassapoyiannis and Voutsinas). Extensive validation of the code has been also performed in the framework of the HeliNovi project where aerodynamic and structural loads, wake velocities and elastic deflections have been compared to tunnel measurements on a BO105 helicopter model (Dieterich et al., 2005).



## 8 Results

In the following section, the effectiveness of the iteration procedure and the estimation of the relaxation factor are demonstrated for a horseshoe vortex. Then in Sect. 8.2 the unsteady induction predicted by the coupled near and far wake model is compared with results from an unsteady BEM model and the free wake code described in Sect. 7. Pitch steps and prescribed vibrations

of the blades of the NREL 5 MW reference turbine are investigated.

### 8.1   Iteration procedure

To illustrate the efficiency of the iterative implementation, induction buildups for a simplified case are shown in Fig. 11. The simple test case is a wing with a span of 0.3 m and a constant bound circulation, so that only two vortices with opposite vortex strength are trailed at the edges. To use the NWM and ensure parallel flow, the wing is modeled as the only aerodynamic

section at the end of a 10 km long blade. The free stream velocity is 70 m/s. At t=1 s, the geometric AOA of the wing with a symmetrical profile is increased from 0 to 5 degrees within 0.02 s. The lift coefficient is $2\pi\alpha$, the chord 1 m. The left side of Fig. 11 shows the induction buildup for different time steps without iterating, the right side shows the effect of the iteration procedure. Both the overshoot of the induction for a time step of 0.002 s and the oscillations for a time step of 0.02 s are reduced by the iteration procedure.

The relaxation factors estimated as proposed in Sect. 5.2 are compared with the lowest stable relaxation factors obtained by trial-and-error in Fig. 12 for the NREL 5 MW reference turbine operating at 8 and 25 m/s wind speed in uniform inflow. The comparison in Fig. 12 shows that the estimated relaxation factor is conservative, but the safety margin towards unstable computation is smaller in the 25 m/s case.

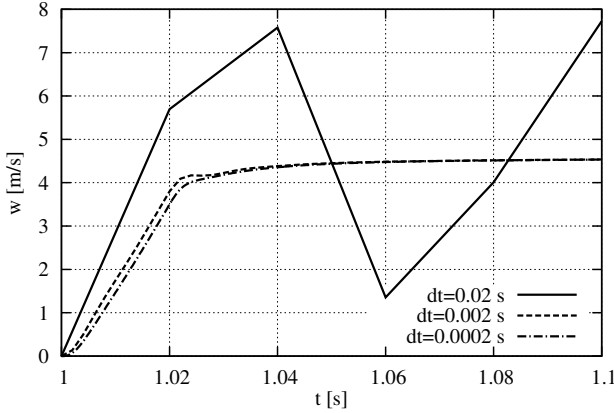
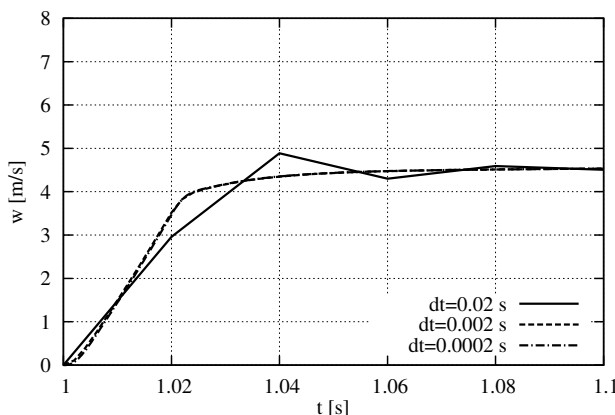

**Figure 11.** Buildup of the downwash for a horseshoe vortex depending on the time step. The NWM tends to be unstable (left) but can be stabilized by iterating to convergence of the downwash (right).





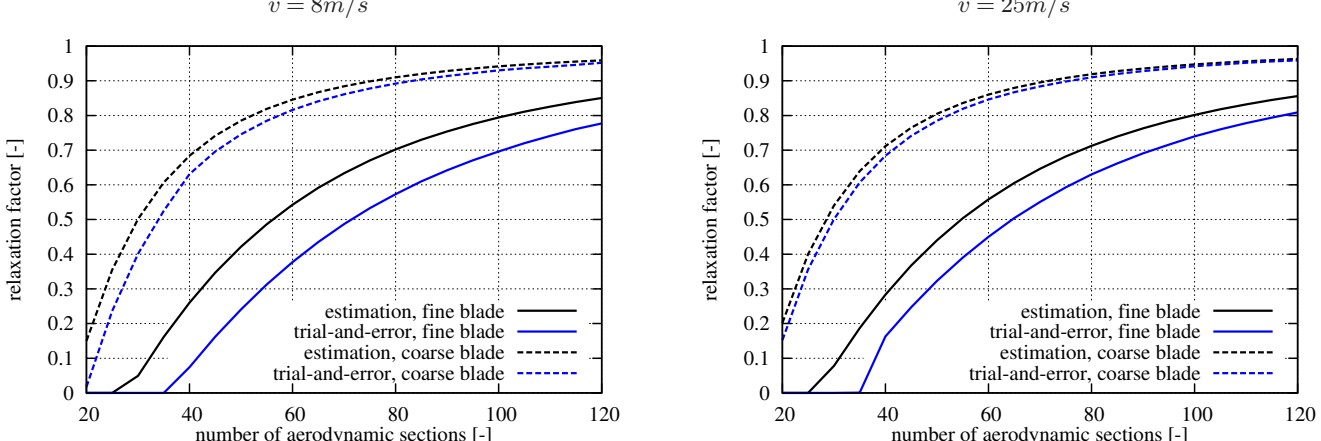

**Figure 12.** Estimated relaxation factor compared with the lowest stable relaxation factor from trial-and-error depending on the number of aerodynamic sections. The time step is 0.02 seconds. The estimated relaxation factors, Eq. (31), are conservative and the influence of the refined blade geometry is captured.

## 8.2 Comparison of the coupled model with a BEM model and a free wake panel code

The predicted force responses to pitch steps (Sect. 8.2.1) and blade vibrations (Sect. 8.2.2) are investigated in the following. All computations use the refined blade model shown in Fig. 6. The blade has been discretized using 40 radial aerodynamic stations in the BEM based codes, with corresponding 41 vortices in the coupled model trailed from root, tip and in between

stations. For the lifting surface free wake simulations, the blade has been discretized using 35 span wise and 11 chord wise grid lines. Compared to the the faster models, the resolution was mainly reduced close to the blade root.

### 8.2.1 Pitch steps

Pitch steps with stiff blades have been performed, where the NREL 5 MW reference turbine is operating at a wind speed of 8 m/s and a rotation speed of 9.2 rpm. The turbine starts with blades that are pitched by 5 degrees to feather. The time steps

are 0.054 seconds (120 steps per revolution) for GENUVP and 0.05 seconds for the other models. After 60 seconds simulated time, the blades are pitched to 0 degrees at constant pitch rate in 1 or 4 seconds. The forces are normalized to compare the dynamics of the pitch response, such that the force before the pitch step is 0 and the force 45 seconds after the pitch step is 1.

Figure 13 show the axial force response at a position at mid-blade and close to the blade tip for the fast pitch step. The free wake code predicts a slower force response during the pitch step than the BEM model. The results of the coupled model

during the pitch step lie in between the other codes. In the free wake code results, some oscillations due to the changing wake geometry are present after the pitch step, especially at the mid blade section. These oscillations make it difficult to judge if the BEM model or the coupled model are predicting the overshoot closer to the free wake code, the results of which are in between the two.





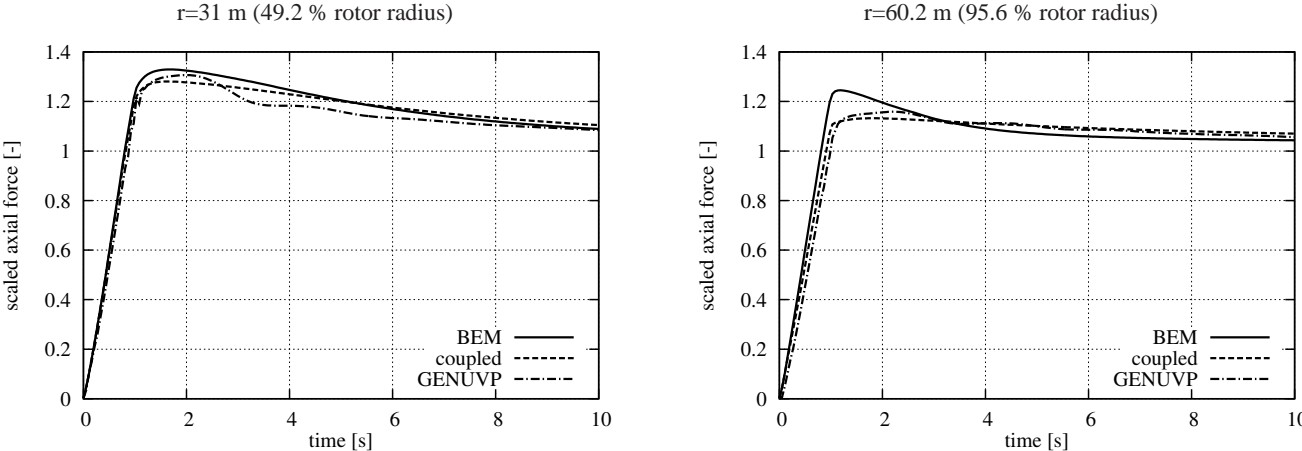

**Figure 13.** Scaled axial force at different radial positions during and after a pitch step by 5 degrees in 1 s.

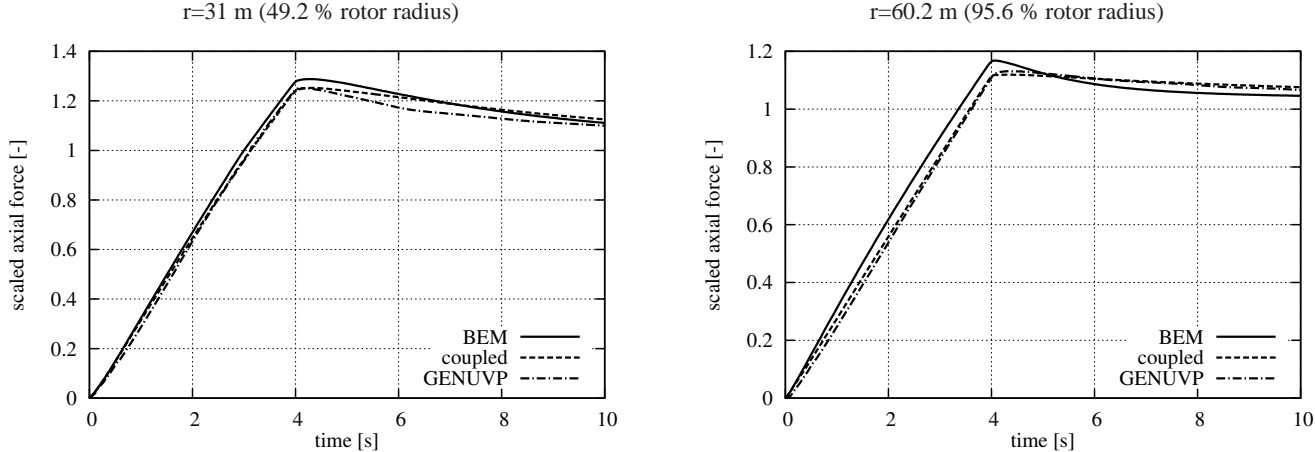

**Figure 14.** Scaled axial force at different radial positions during and after a pitch step by 5 degrees in 4 s.

The results of the slower pitch step in Fig. 14 show less oscillations of the free wake code results. In this case, the predicted results from the coupled model clearly agree better with the free wake code than the BEM results, both on the slope during the pitching motion and on the predicted overshoot.

Axial force distributions for a partial pitch comparison at 8 m/s are shown in Fig. 15. In this case, only the outer half of the
5  blade is pitched from five to zero degrees during 1 second. As shown in the left plot of Fig. 15, the load smoothing effect of the cross sectional coupling at the mid blade due to the trailed vorticity is predicted by both the coupled aerodynamics model and the free wake code to a similar degree. In the right plot of Fig. 15, the time history of the axial force is compared at 34.3 % radius. The constant force predicted by the BEM model on this non-pitching part of the blade is not included in this comparison. The overshoot is underpredicted by the coupled model by around 40%. The induction predicted by the coupled model stops



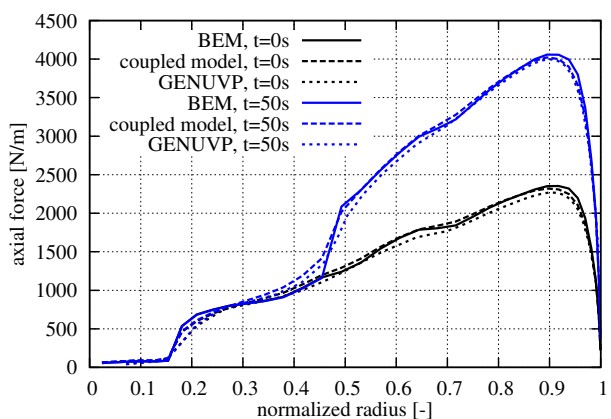 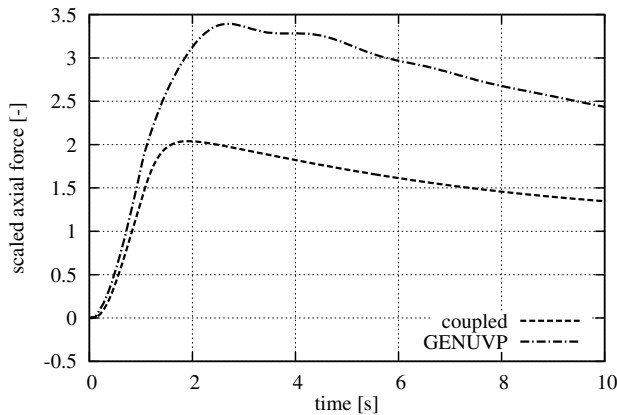

**Figure 15.** Left plot: Force distribution before the pitch step and 50 seconds after.

Right plot: Time history of the axial force comparing coupled model and GENUVP at 21.6 meter radius (34.3 % rotor radius). The BEM gives a constant force.

increasing after 1.6 seconds, corresponding to a quarter revolution at 9.2 rpm, while the force predicted by GENUVP continues to increase. Figure 15 thus illustrates that the coupled model can predict a change in loading that can not be computed based on BEM theory, but the restriction to a quarter revolution is limiting in this case. The difference in the overshoot prediction at 21.6 meters radius amounts to roughly 70 N/m.

**8.2.2 Prescribed vibrations**

The aerodynamic response to blade vibrations is investigated for normal operation at 8 and 25 m/s. The corresponding rotor speeds are 9.2 rpm and 12.1 rpm and the pitch angles 0 degrees and 23.2 degrees, respectively. The force response is compared in terms of radial distributions of aerodynamic work during one oscillation, where a positive aerodynamic work corresponds to a positive aerodynamic damping of the vibration. The mode shapes are chosen as the first and second structural mode

shapes of the NREL 5 MW reference turbine blade at stand still, cf. Fig. 16. To simplify the comparison, the vibrations have been prescribed as collective in-plane or out-of-plane vibrations. The frequencies, amplitudes and time steps used for the computations are shown in Table 1, as well as the modal masses that are used for damping estimations. For the first modes only results at the larger amplitudes are shown in the following. Investigating the results at the smaller amplitudes leads to the same conclusions.

In the BEM and coupled model, the blade section velocities due to the vibrations are added to the relative wind speed. The deflection of the blade and the resulting change of the section positions have been neglected because the amplitudes are small compared to the blade radius.

To distinguish the effects of shed and trailed vorticity, the following comparisons include a BEM computation with a disabled shed vorticity model. The aerodynamic work during out of plane motion according to the first flap mode shape is shown in

Fig. 17 for 8 m/s and 25 m/s. The work integrated over the blade is overpredicted by the BEM model (dashed black line) by



| Mode | f. [Hz] | m. mass [kg] | amp. [m] | | dt [s] |
|---|---|---|---|---|---|
| first flap | 0.66 | 905 | 0.25 | 0.5 | 0.036, 0.028 |
| first edge | 1.0 | 1480 | 0.25 | 1.0 | 0.025 |
| second flap | 1.82 | 594 | 0.25 | | 0.01 |
| second edge | 3.2 | 793 | 0.25 | | 0.01 |

**Table 1.** Modal parameters and time steps prescribed in the work comparison. The time steps for the first flap were chosen as 180 steps per revolution and thus depend on the rotor speed.

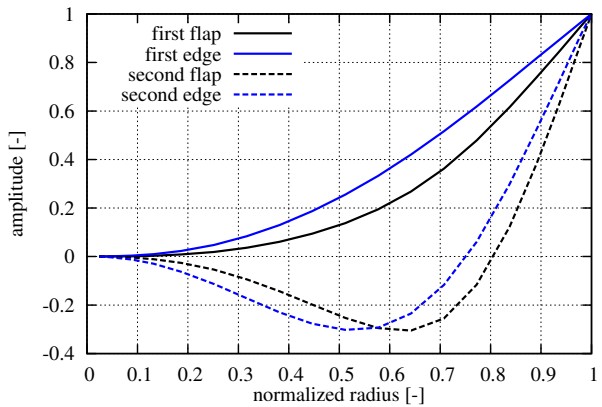

**Figure 16.** Mode shapes used in the work computations, which are simplified to be purely in-plane or out-of-plane deflections.

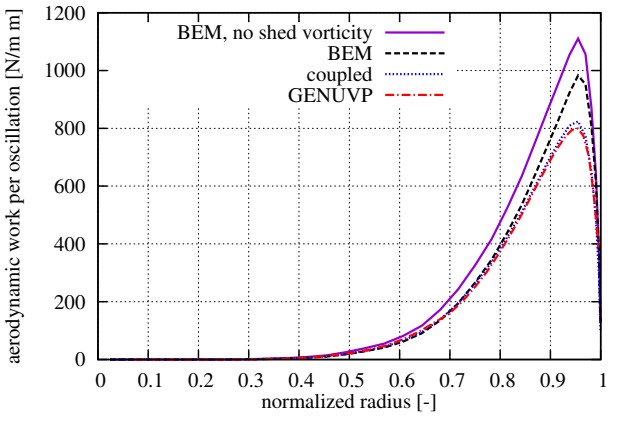

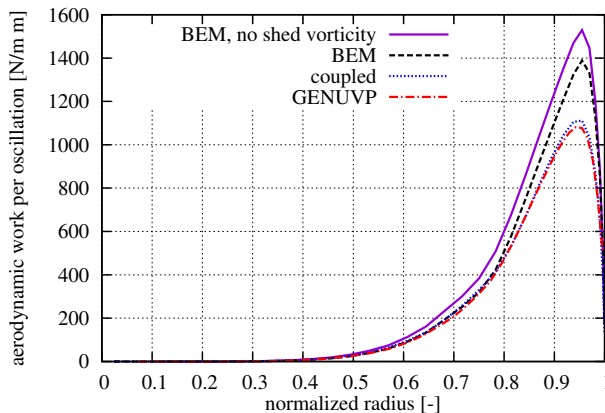

**Figure 17.** Aerodynamic work per oscillation of first flap motion at 8 m/s (left) and 25 m/s (right) wind speed with an amplitude of 0.5 m.

about 10% compared to the free wake code in both cases. If the three BEM based results are compared, it shows that the trailed vorticity (dotted blue line versus dashed black line) has an influence of the same order of magnitude as the shed vorticity (solid purple versus dashed black line). The trailed vorticity effects, are more important close to the tip vortex, while the influence of the shed vorticity extends across the whole blade. The results of the coupled model are very close to the free wake code results, but deviate slightly towards higher work. The influence of the trailed vorticity behind the other two rotor blades, which is not included in the NWM, on the vibration response is found to be small compared to the influence of the wake of the blade itself in normal operation.

The in-plane vibrations at 8 m/s are almost parallel to the inflow and the drag forces contribute much more to the work than in the other cases. To simplify the problem, drag has been excluded from the aerodynamic work computations presented in Fig. 18. Further, the lift gradient has been assumed as $2\pi$. In the left plot of Fig. 18, the quasi steady angle of attack is used in the unsteady airfoil aerodynamics model, cf. Equations (8) to (11). The agreement in this case is poor. In the right plot, the zero




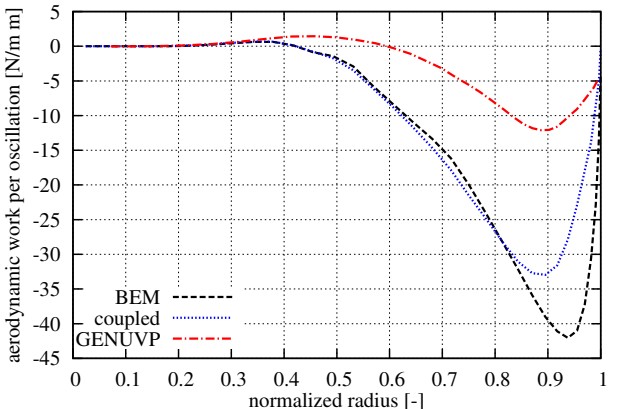
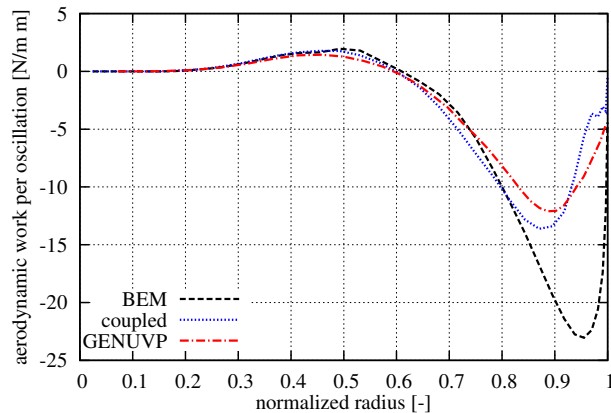

**Figure 18.** Aerodynamic work per oscillation of first edge motion at 8 m/s at an amplitude of 1 m. Drag has been excluded, and $C_L = 2\pi\alpha$. In the left plot, $\alpha_{QS}$ is used in the unsteady airfoil aerodynamics model in both coupled model and BEM, cf. Equations (8) to (11). In the right plot, the unsteady airfoil aerodynamics model uses $\alpha_{QS} - \alpha_0$, cf. Eq. (20), improving the agreement with the free wake code results.

lift angle due to camber is included in the quasi steady angle of attack, cf. Eq. (20). This approach leads to a much improved result for the BEM based codes and good agreement of the coupled model and GENUVP.

With the airfoil polars of the NREL 5 MW reference turbine the agreement between the codes is not as good in the 8 m/s case, see the left plot of Fig. 19. However, the coupled model produces results much closer to the free wake code close to the blade tip than the BEM model. At 25 m/s, where the work is predominantly due to the vibration component perpendicular to the inflow as a result of blade pitch, the coupled near and far wake model agrees similarly well with the free wake code as in

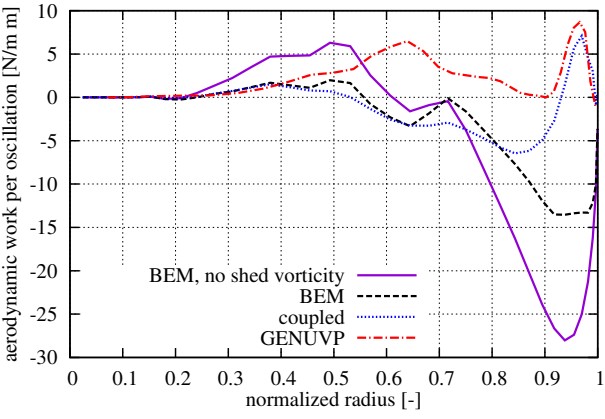
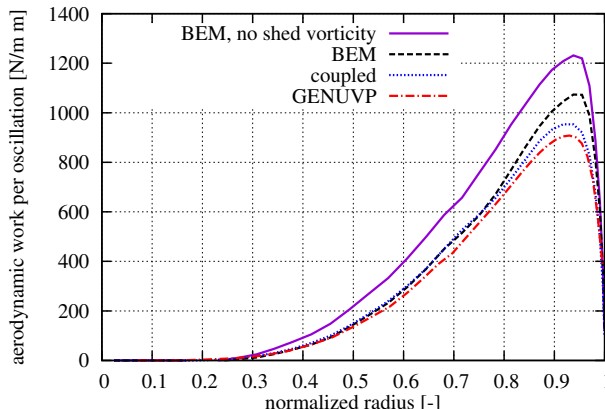

**Figure 19.** Aerodynamic work per oscillation of first edge motion at 8 m/s (left) and 25 m/s (right) wind speed with an amplitude of 1.0 m.





the cases with out-of-plane vibrations discussed above. The shed vorticity effects on the in-plane vibrations are larger than on the out-of-plane vibrations due to the higher frequency and the larger relative velocity variations.

Figure 20 shows results for the second modes. The BEM model results compare similarly well with the GENUVP results as for the first modes. Because the frequencies of the second modes are higher the shed vorticity model is more important in
5 these cases. The importance of the trailed vorticity model at higher frequencies does not increase by the same amount, because the higher reduced frequencies affect the buildup of the unsteady bound circulation, cf. Eq. (18). The coupled model results are closer to the free wake results than the BEM results in all cases, but as opposed to the comparisons above, the coupled model is underestimating the effects of the dynamics of the tip vortex. Further inboard the free wake code predicts slightly lower aerodynamic work in the 25 m/s case than the BEM model, which can't be seen in the coupled model results. Also the
10 agreement of the coupled model and GENUVP is worse in the edgewise case than in the flapwise case at 25 m/s, which has not

**Figure 20.** Aerodynamic work per oscillation of second flap (top) and edge (bottom) motion at 0.25 m amplitude at 8 m/s (left) and 25 m/s.



been seen to that extend for the first edgewise cases, cf. Figures 17 and 19. A reason for this might be that the second edgewise case is computed with fewer time steps per period of oscillation to limit the computational cost.

For easier evaluation of the force response differences, the aerodynamic work can be expressed in terms of a damping ratio of a respective blade mode. Because the computations have been based on prescribed purely in-plane and out-of-plane structural mode shapes, these dampings do not correspond to any aeroelastic blade modes. For a single degree of freedom system with the modal mass $m$ and frequency $f$, given in Table 1, the damping ratio $\xi$ and logarithmic decrement $\delta$ are:

$$\xi = \frac{W_{aero}}{8\pi^3 A^2 f^2 m} = \frac{1}{\sqrt{1 + \left(\frac{2\pi}{\delta}\right)^2}} \tag{36}$$

where $A$ is the amplitude and $W_{aero}$ the aerodynamic work per oscillation period. The estimated logarithmic decrements according to Eq. (36) corresponding to the first flap motion at 8 m/s with an amplitude of 0.5 m, cf. Fig. 17, are 334 % for the BEM results, 300 % for the coupled model and 292 % for the free wake code results. Flapwise modes are highly damped and thus these changes of the damping will not significantly alter the blade fatigue loads. On the other hand, lower aerodynamic damping of flapwise blade motion will correspond to a lower aerodynamic damping of tower fore-aft motion and might thus lead to increased tower fatigue loads. It is expected that this lower aerodynamic damping is balanced to some degree because the near wake effects reduce the aerodynamic excitation due to atmospheric turbulence.

Aerodynamic damping estimations for the first in-plane vibrations at 8 m/s at 1 m amplitude are shown in Table 2. The damping has been estimated in four cases, which differ in the airfoil polars and the modeling of the camber effect on the unsteady airfoil aerodynamics. Comparison of the first two cases of Table 2 shows that the induced drag caused by airfoil camber in the shed vorticity modeling results in a damping of roughly 0.7 % logarithmic decrement. According to the BEM and coupled model results in case (3) and (4) the airfoil drag increases the logarithmic decrement by about 0.3 %. The trailed vorticity decreases the absolute value of the damping by roughly 0.14 % log. dec. Further, comparing columns (2) and (4), the combined influence of airfoil polars with lift coefficients other than $2\pi$ and drag is close to three times larger in the free wake code computations, which is caused by the different unsteady drag modeling. The small differences in estimated logarithmic decrement can have an impact on loads and stability computations for edgewise modes with a very low aeroelastic damping.

In the out-of-plane prescribed vibration cases investigated, the trailed vorticity reduces the aerodynamic work. Further a previous study by Pirrung et al. (2014) showed that the trailed vorticity effects will delay the onset of flutter towards higher rotor speeds. This is in agreement with findings on the influence of shed vorticity, which leads to both a decrease of the flapwise damping and increased flutter speeds of a vibrating 2D blade section (Hansen, 2007).

| | (1) $C'_L = 2\pi$, $C_D = 0$, no camber | (2) $C'_L = 2\pi$, $C_D = 0$ | (3) NREL $C_L$, $C_D = 0$ | (4) NREL $C_L$ and $C_D$ |
|---|---|---|---|---|
| BEM | -1.13 | -0.41 | -0.52 | -0.24 |
| Coupled model | -1.00 | -0.27 | -0.37 | -0.1 |
| GENUVP | - | -0.25 | - | 0.22 |

**Table 2.** Estimated logarithmic decrements [%] corresponding to the aerodynamic work of first in-plane vibrations at 8 m/s, based on the results at an amplitude of 1 m.



## 9 Conclusions

In this paper, several modifications of a coupled model consisting of a trailed vorticity model for the near wake and a BEM-based model for the far wake have been presented and validated. Results from the coupled model are compared to free wake panel code and a BEM model to evaluate the benefits and limitations of the added trailed vorticity modeling.

It has been shown that the acceleration of the model by reducing the number of exponential functions in the trailing wake approximation from two to one is possible with negligible effect on the results. The approach presented here does not change the steady results predicted by the NWM.

An iteration scheme to stabilize the model has been presented. It applies a relaxation factor that is computed dynamically based on the blade discretization and the operating point of the turbine. To evaluate the computed relaxation factors, minimum

necessary relaxation factors have been determined by trial-and-error and the estimated factors are found to be conservative. The iterative process enables stable computations without the need for very small time steps and reduces oscillations of the near wake induction.

The 2D shed vorticity modeling, based on thin airfoil theory, has been extended by including the unsteady effects on the bound circulation. Further it has been found that it is necessary to include airfoil camber in the modeling of the influence of

varying inflow velocity on the dynamic angle of attack to obtain good results if the direction of vibration is close to parallel to the inflow direction.

A comparison of pitch step responses of the NREL 5 MW reference turbine using the coupled near and far wake model, a BEM model based on the aerodynamics model in HAWC2 and the free wake panel code GENUVP has been presented. The trailed vorticity modeling in the coupled model gives results closer to the free wake code than the BEM model during the

pitching motion, and for a slow pitching rate a clear improvement is seen in the computation of the overshoot. Fast pitch rates resulted in oscillations due to the motion of the wake in the free wake code, which could not be achieved in the coupled model due to the prescribed wake assumption. The response to a partial pitch of the outer half of the blade demonstrated the cross sectional aerodynamic coupling, which will have an influence on the load distribution in the presence of trailing edge flaps.

The coupled model agreed better than the BEM model with the free wake code in all prescribed vibration cases investigated.

The main improvement due to the trailed vorticity is found close to the tip of the blade, even in case of the higher modes investigated. The work response to the edgewise vibrations has been found to be difficult to model if the direction of vibration is close to parallel to the inflow direction. The results in this case compare much better if no drag forces are computed. If drag is included, the coupled model still compares well with the free wake code close to the blade tip, but there are larger deviations in the results of all models further inboard. In general, the simulations agreed better for out-of-plane vibrations than in-plane

vibrations.

The implementation of the coupled near and far wake model presented here delivers promising results and will be further investigated and validated against computational fluid dynamics results and measurements in future work. In particular the more accurate prediction of aerodynamic work for edgewise vibrations is considered to be important for stability analyses and load predictions due to the low aeroelastic damping typically associated with these vibrations.



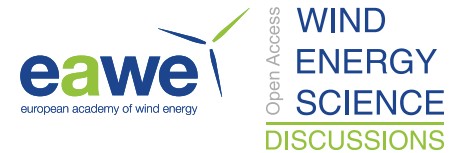

## Nomenclature

| Symbol | Units | Description |
| --- | --- | --- |
| $A$ | [m] | Amplitude of vibration |
| $A_1, A_2$ | [-] | Dynamic inflow weighting factors |
| $A_{\Gamma,1}, A_{\Gamma,2}, A_{\Gamma,3}$ | [-] | Coefficients in bound circulation step response |
| $A^{\star}$ | [-] | Coefficient of accelerated near wake model |
| $a_{FW}$ | [-] | Far wake induction factor |
| $a_{ref}$ | [-] | Induction factor according to BEM-polynomial |
| $b_{\Gamma,1}, b_{\Gamma,2}, b_{\Gamma,3}$ | [-] | Dimensionless time constants in bound circulation step response |
| $C_D$ | [-] | Drag coefficient |
| $C_L$ | [-] | Lift coefficient |
| $C_L'$ | [-] | Gradient of lift coefficient with respect to angle of attack |
| $C_T$ | [-] | Thrust coefficient |
| $c$ | [m] | Chord length |
| $D_{X,s,v}$ | [m$^{-1}$] | Slowly decaying component of induced velocity due to infinitely long vortex arc $v$ with vortex strength 1 at section $s$ |
| $D_{Y,s,v}$ | [m$^{-1}$] | Fast decaying component of induced velocity due to infinitely long vortex arc $v$ with vortex strength 1 at section $s$ |
| $dw$ | [m/s] | Induced velocity due to infinitesimal vortex element |
| $dw_0$ | [m/s] | Induced velocity due to infinitesimal vortex element starting at the blade |
| $F_{aero}$ | [N/m] | Aerodynamic forces per unit radius |
| $F_{ip}$ | [N/m] | In-plane component of the aerodynamic forces per unit radius |
| $F_{oop}$ | [N/m] | Out-of-plane component of the aerodynamic forces per unit radius |
| $F$ | [-] | Tip loss factor |
| $f$ | [Hz] | frequency of vibration |
| $f_r$ | [-] | Relaxation factor |
| $k$ | [-] | reduced frequency |
| $k_{FW}$ | [-] | Coupling factor |
| $L$ | [N/m] | Lift force per unit radius |
| $m$ | [kg] | Modal mass |
| $N_v$ | [-] | Number of vortex arcs trailed from a blade |
| $u_1, u_2$ | [m/s] | Components of time filtered far wake induced velocity |
| $u_{FW,dyn}$ | [m/s] | Time filtered far wake induced velocity |
| $u_{FW,QS}$ | [m/s] | Quasi steady far wake induced velocity |
| $u_{tot}$ | [m/s] | Total induced velocity due to the combined near and far wake |



| Symbol | Units | Description |
|---|---|---|
| $u_\infty$ | [m/s] | Free wind speed |
| $T_0$ | [s] | Time constant for unsteady airfoil aerodynamics model |
| $T$ | [s] | Period of oscillation |
| $\Delta t$ | [s] | Time step |
| $v_{ip}$ | [m/s] | In-plane component of the blade section velocity |
| $v_{oop}$ | [m/s] | Out-of-plane component of the blade section velocity |
| $v_{r,in-plane}$ | [m/s] | In-plane component of relative velocity |
| $v_r$ | [m/s] | Relative velocity |
| $W_{aero}$ | [Nm/m] | Aerodynamic work during one period of oscillation per unit radius |
| $W_{s,v}$ | [m/s] | Induced velocity due to vortex arc $v$ at section $s$ |
| $W_s$ | [m/s] | Induced velocity due to all vortex arcs at section $s$ |
| $X_{s,v}$ | [m/s] | Slowly decaying component of induced velocity due to vortex arc $v$ at section $s$ |
| $x_1, x_2$ | [m/s] | Components of effective angle of attack |
| $x_{\Gamma,1}, x_{\Gamma,2}, x_{\Gamma,3}$ | [m$^2$/s] | Components of bound circulation |
| $Y_{s,v}$ | [m/s] | Fast decaying component of induced velocity due to vortex arc $v$ at section $s$ |
| $\alpha_{QS}$ | [rad] | Geometric angle of attack |
| $\alpha_{QS,camber}$ | [rad] | Geometric angle of attack relative to zero lift angle |
| $\alpha_0$ | [rad] | Zero lift angle |
| $\alpha_{eff}$ | [rad] | Effective angle of attack |
| $\beta$ | [rad] | Angle a blade rotated since a vortex has been trailed |
| $\Delta\beta$ | [rad] | Angle a blade rotates in one time step |
| $\delta$ | [-] | Logarithmic decrement |
| $\Gamma_{QS}$ | [m$^2$/s] | Bound circulation |
| $\Delta\Gamma$ | [m$^2$/s] | Trailed vortex strength |
| $\Omega$ | [rad/s] | Rotor speed |
| $\omega$ | [rad/s] | angular velocity |
| $\Phi$ | [-] | Geometric parameter determining how fast the influence of a trailed vortex element decays |
| $\Phi^\star$ | [-] | Geometric parameter of accelerated near wake model |
| $\tau_1, \tau_2$ | [s] | Dynamic inflow time constants |
| $\tau$ | [-] | Dimensionless time in bound circulation step response |
| $\xi$ | [-] | Damping ratio |



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
