# Peer review of "Comparison of a Coupled Near and Far Wake Model With a Free Wake Vortex Code"

_Wind Energy Science, 2016_

## Referee Comment (RC1) · S. Voutsinas (Referee) · 14 Jun 2016

**Comparison of a Coupled Near and Far Wake Model With a Free Wake Vortex Code**

Georg Pirrung, Vasilis Riziotis, Helge Madsen, Morten Hansen, and Taeseong Kim

**General Comments**

The specific research aims at formulating a fast but *advanced comprehensive* aerodynamic model for wind turbines. It combines a vortex based near wake with a BEM far wake modeling. In this respect it may be viewed either as a simplification of the lifting line model or as an improvement of BEM modeling. The simplification of the lifting line part consists of retaining only one quarter of the trailing vortices. For the remaining (far wake) part of the induction BEM modeling is used. Otherwise the concept of evaluating an effective angle of attack along with an effective relative velocity and then using look-up tables for the loads remains – as also done in full vortex methods. Finally in order to keep the cost low a number of numerical approximations are introduced.

The above general description applies to the original idea of Beddoes regarding the near wake and previous works by the authors on the far wake modeling. In this regard, the present contribution looks into the details of the model and ways to improve its performance in terms of numerical stability, cost and applicability.

**Specific comments on the modeling**

1. While keeping one quarter of the trailing vortices can be viewed as a compromise in order to have low cost, a comment should be added. For a 3 bladed wind turbine this choice prevents the blade to sense the wake of the preceding blade which may play a role at low wind speeds.

2. The split in X and Y components of W in eq (2) should be explained

3. Adopting eq (1) suggests that the expansion of the wake should be taken into account indirectly by adjusting the constants. Was this aspect considered?

4. In unsteady conditions the wake will also include radial vortex elements. Eq (15) intends to take into account the effect on the bound vorticity. In this eq the normal to the lifting line velocity $v_r$ regulates the result. Does $v_r$ correspond to the quasi-static set-up?

5. Since the near wake elements carry constant circulation, in unsteady conditions Kelvin's theorem will be violated. Was the option of switching to linear circulation distributions considered?

6. The CL slope of $2\pi$ originates from thin airfoil theory. This is not the case with thick airfoils. It would be interesting to check whether switching to different slope would improve the results.

7. The implementation contains several constants. The question is whether they depend on the blade design or they can be regarded as universal

8. It is clear that the main concern is to keep the cost low. How does the run time compare with a pure BEM simulation?

**Specific comments on the results**

1. Figure 12. If the simulations take into account the proper pitch, why is the margin to instability smaller at 25m/s? Could that be due to the fact that at 25m/s the wake is convected faster than in the case of 8m/s?

2. Full pitch steps: The discussion is based on two metrics: the slope of the axial force during the step and the post-step shape of the variation. With respect to the slope, the discussion could also include the angle of the spiral which should be bigger at more inboard stations. This may offer an additional explanation why BEM is in better agreement at r=31m for both steps as compared to the far outboard section. Also the fact that the post step oscillations seen in the free-wake model do not appear in the coupled model simulations may suggest that the exponential approximations of the dynamics of the circulation (and angle of attack) act as filters in particular as regards the radial wake vortex elements (See previous point 4).

3. Partial pitch stepping: Although the load variation at r/R=0.45 is smooth in the coupled and free wake results as compared to BEM, the specific characterization may be misleading – smoothening is often connected to numerical damping or filtering while in the present case there is a physical mechanism that is driving this difference. The trailing vortex shed at the turning point will increase the incidence towards the root and decrease it on the other side.

4. Aerodynamic work computations: The results clearly indicate that BEM modeling is conservative (mainly with reference to Case 2 in Table 2 for which the results are not subjected to dynamic inflow modeling specifics). However the damping in the edgewise mode remains negative. Did the simulations only considered the rotor and how the specific result can be interpreted?

**Concluding remarks**

Improving BEM modelling is an never ending process. To my opinion including the near-wake model proposed by Beddoes is a valid research line. In this respect, the paper formulates specific improvements of this approach. This brings the specific modelling at a more mature level in particular with respect to stability analysis. The improvement in comparison to BEM modelling is supported by the results presented in the paper.

A lot of effort was put in keeping the cost low, so an average estimate of the cost should be added. Another point concerns the constants that are set. Are they universal? And if not, which of them could be subjected to calibration according to the authors' experience?

---

## Referee Comment (RC2) · Anonymous Referee #2 · 25 Jun 2016

**General comments The paper presents several innovations for the implementation of coupled near-wake/far-wake model. The model is described in details and compared to a free wake vortex code. The method appears to improve the BEM model at a low computational cost.**

The new model accounts for the difference in time scales of the far wake model depending on the operating conditions and introduce advanced unsteady aerodynamics corrections. The authors address convergence issues and provide an analytical expression for the determination of the relaxation factor which is essential to reach convergence. Computational time is improved by using this "optimal" relaxation factor and by reducing the number of exponential components of the indicial function. The innovations presented in the paper appear to improve previous implementations. The comparison to a free wake vortex code is relevant and the investigations related to the

aerodynamic damping is of great importance for turbine fatigue loads.

As a general comment, the inherent differences between the two models should be addressed (either in a discussion paragraph, or by adding extra-investigations), in particular: lifting-line versus lifting-surface approach, rigid curved filaments with no expansion versus free particle, differences in dynamic stall implementations. Each of these "model" differences will have an impact on the results, but the extent of this impact is uncertain. The paper could have even greater value if the impact of this modeling choices were investigated: using a lifting formulation in GENUVP, by deactivating the dynamic stall in both models, etc.

Other comments are listed below.

**Specific Comments p2 l8-9 and p2 l22-26: It seems you are giving results in these sentences. If they are results from this paper, you could consider moving them outside of the introduction part, or remove them completely if they are mentioned further in the document.**

p4 l18: Could you elaborate more on the correction for the helix angle? The correction is not part of 3a 3b, is it? The sentence seems to say the correction is applied "on" the helix angle. Does the corrected model still assumes that the vorticity lays in the rotor plane? These were questions that came to me when reading this sentence. They are answered in the given reference but some details could help the reader at that point. (In fact you address this later on, in section 3. But the reader can be confused when it is introduced in section 2).

p4 equation (4): This is understood as a sum of vectors. In this paper, a visual distinctions between vectors and scalars could help the reader.

p4 l20 and l24-25: a reference is needed for the BEM model implementation and the a-CT relation used.

p4 l25-28: The description was hard to follow for me. When it is written "to account

for the near wake induction" is it meant in the sense "to compute a 'reduced far-wake' induction which does not include the effect of the near wake induction"? My own re-formulation is poor, but could you reformulate the lines 25-28 to guide the reader? Also, could you add a reference or justification to support the removal of the tip-loss factor? Further, though understood, it is probably partially true that a_FW was computed without tip-loss corrections, since the thrust coefficient was probably computed using loads on the blades which in turn were determined included a tip-loss factor. If this is correct, could you comment on that, maybe add an equation to clarify what is meant here?

p9 l9: The author uses a cosine distribution which has indeed some numerical advantages for "elliptic" wings. Number of sections up to 100 were used in this study. Is the model stable above this number of sections? Has the author tried 200 or more sections? Could the elliptic distribution introduce convergence problems or numerical issues due to the small radial elements at the extremities? Have you investigated linear distributions and observed loss of performances (more iterations/computational time, decreased accuracy at the extremities, problems of convergence)? Could the linear distribution then out-perform the elliptic distribution for large number of sections? These questions are slightly rhetorical but your experience on the topic could be beneficial for the community.

p14 l11: It is not obvious what the "hybrid wake" approach is referring to. Does it refer to the way the far wake influence is computed? Or the combination of dipoles and particles?

p15 figure11: The figure demonstrates the effect of the iterative procedure. It could add value to also compare the results with other studies of the acceleration of a flat plate from rest. It can give insights into the validity of the physics and time constants of this model (which was mainly tuned for rotors).

p17+: Oscillations are observed in the results of the GENUVP code. It could add value to the analysis to consider a case where the dynamic stall model was removed

from both codes in order to compare the build-up of inductions from both codes. The empiricism from the dynamic stall models introduces an extra level of complexity and thus more uncertainty on the source of the observed differences.

p19 l15: How are the vibrations prescribed for the GENUVP code?

**Technical corrections p1 l20: I personally had trouble reading the second part of the sentence. Maybe you could replace "far wake computation is using", by "the far wake contribution is computed using"?**

p2 l17: Consider splitting the sentence which is a bit long.

p14 l13-14: The sentence "If the relaxation factor..." seems incomplete.

p14 l16: The term "free flow velocity" may be reformulated to "local velocity" to avoid confusion with the free stream velocity.

p15 l1-3: The sentence is difficult to read, could it be reformulated? Also, consider replacing "local flow angle of attack" to "angle of attack" (or "3D angle of attack" or "local angle of attack").

p15 l9-11: Reformulation of these lines could help the reading.

p23 l21: Maybe "and zero drag" is meant. The structure of the sentence is hard to follow.

---

## Author Comment (AC1) · 30 Jul 2016

*Specific comments on the modeling*
*1. While keeping one quarter of the trailing vortices can be viewed as a compromise in order to have low cost, a comment should be added. For a 3 bladed wind turbine this choice prevents the blade to sense the wake of the preceding blade which may play a role at low wind speeds.*

- For the fast dynamic response we are mainly concerned about the most important effects from the blade itself. In the GENUVP vibration results there was only a very small influence from the other blades visible when more periods of oscillation were simulated, but instead the work in one oscillation stayed almost constant.
- This is also consistent with the 2D shed vorticity computation, where the shed wake from other blades is ignored.
- An option to include the aerodynamic interaction between blades on the far wake BEM grid (which is likely the reason for the post pitch step oscillations in the free wake code in Section 8.2.1) will be investigated in the future.

*2. The split in X and Y components of W in eq (2) should be explained*

- The X and Y components contain the contributions according to the two exponential functions in eq 1.
- There is only one component if the acceleration method proposed in Section 6 of the article is used.
- These explanations will be included in the revised article.

*3. Adopting eq (1) suggests that the expansion of the wake should be taken into account indirectly by adjusting the constants. Was this aspect considered?*

- At low wind speeds where the wake expands it is only convecting quite slowly, so after 1/4 rotation we are still close to the rotor plane and we expect that this effect on the near wake part is negligible.

*4. In unsteady conditions the wake will also include radial vortex elements. Eq (15) intends to take into account the effect on the bound vorticity. In this eq the normal to the lifting line velocity v_r regulates the result. Does v_r correspond to the quasi-static set-up?*

- The relative velocity is updated in each time step (as in the dynamic stall / shed vorticity model).

*5. Since the near wake elements carry constant circulation, in unsteady conditions Kelvin's theorem will be violated. Was the option of switching to linear circulation distributions considered?*

- It has been considered, however
    - the iteration procedure makes sure that the constant circulation trailed during a time step is not unrealistically high (which might otherwise lead to instability).

- because the model is fast and used in an aeroelastic code, the chosen time steps will be in the order of 1 to 2 degrees blade rotation per time step, so they are small enough to ensure good results with constant circulation
- implementing a linear distribution has so far been found difficult due to the approximations in the model

*6. The CL slope of 2π originates from thin airfoil theory. This is not the case with thick airfoils. It would be interesting to check whether switching to different slope would improve the results.*

- the $2\pi$ slope is only used in the approximation procedure for the relaxation factor (Section 5.2)
- the 2D unsteady airfoil aerodynamics (dynamic stall) model uses the actual slope of the airfoil
- the quasi steady circulation as basis for the unsteady circulation buildup is depending on the quasi steady lift coefficient from the airfoil polar
  - therefore the trailed vortex strength is also based on the lift coefficient from the airfoil polar
- the influence of using thick airfoil based time constants in the indicial functions for the 2D dynamic stall has been found to be small in [Bergami, L., Gaunaa, M. and Heinz, J. (2013), Indicial lift response function: an empirical relation for finite-thickness airfoils, and effects on aeroelastic simulations. Wind Energ., 16: 681–693. doi:10.1002/we.1516]

**7. The implementation contains several constants. The question is whether they depend on the blade design or they can be regarded as universal**

- The constants are either based on thin airfoils (2D models) or on the wake geometry (near wake model)
- it is shown how the constant Φ changes with helix angles in [Pirrung, G. R., Madsen, H. A., Kim, T., and Heinz, J. (2016) A coupled near and far wake model for wind turbine aerodynamics. Wind Energ., doi: 10.1002/we.1969.]
- so far we approximate the blade as a straight lifting line for the modeling of the near wake dynamics. Including prebend/sweep and deformations in the blade model would in principle also change the constants, mainly Φ

**8. It is clear that the main concern is to keep the cost low. How does the run time compare with a pure BEM simulation?**

- The computational effort for the aerodynamics is much higher than in a BEM simulation. In the context of an aeroelastic model, where the structural part is responsible for a large part of the computational effort, the near wake model can be used without a drastic increase in computational time. HAWC2 simulations have been found to run approximately 5-50% slower when the near wake model is enabled compared to BEM aerodynamics. The additional effort depends on the discretization (number of aerodynamic points and distribution), blade geometry and operating conditions, so it is difficult to give a single estimate.

*Specific comments on the results*
*1. Figure 12. If the simulations take into account the proper pitch, why is the margin to instability smaller at 25m/s? Could that be due to the fact that at 25m/s the wake is convected faster than in the case of 8m/s?*

- one assumption is that the computation is done without prior induced velocity
    - in the 8 m/s case where the induction is generally larger this larger induction will in practise help stabilise the computation
    - the large effect of the blade refinement indicates that the instability comes from the tip vortex, which is resolved over more trailed vortices in the fine case.

*2. Full pitch steps: The discussion is based on two metrics: the slope of the axial force during the step and the post-step shape of the variation. With respect to the slope, the discussion could also include the angle of the spiral which should be bigger at more inboard stations. This may offer an additional explanation why BEM is in better agreement at r=31m for both steps as compared to the far outboard section. Also the fact that the post step oscillations seen in the free-wake model do not appear in the coupled model simulations may suggest that the exponential approximations of the dynamics of the circulation (and angle of attack) act as filters in particular as regards the radial wake vortex elements*
*(See previous point 4).*

- The angle of the wake helix is included in the near wake model, so that should not be a reason for disagreements between the codes.
- The post step oscillations occur at 3p frequency and are likely due to the blades passing by the wake of the previous blades after the pitch step.

*3. Partial pitch stepping: Although the load variation at r/R=0.45 is smooth in the coupled and free wake results as compared to BEM, the specific characterization may be misleading – smoothening is often connected to numerical damping or filtering while in the present case there is a physical mechanism that is driving this difference. The trailing vortex shed at the turning point will increase the incidence towards the root and decrease it on the other side.*

- Yes, we clearly see the modeling of a physical mechanism by the free wake and the near wake code.
- A sentence making this more clear will be added in the article.

*4. Aerodynamic work computations: The results clearly indicate that BEM modeling is conservative (mainly with reference to Case 2 in Table 2 for which the results are not subjected to dynamic inflow modeling specifics). However the damping in the edgewise mode remains negative. Did the simulations only considered the rotor and how the specific result can be interpreted?*

- The simulations are purely aerodynamic and computed with a stiff turbine at fixed rotation speed.
- From these results, it appears that both shed and trailed vorticity reduce the absolute value of the aerodynamic work => positive damping and negative damping get smaller

- This is in agreement to the findings published in [Pirrung G, Aagaard Madsen H, Kim T. 2014. The influence of trailed vorticity on flutter speed estimations. Journal of Physics: Conference Series (Online). 524. Available from: 10.1088/1742-6596/524/1/012048], where negative damping of a flutter mode occured at a slightly higher relative speed when trailed vorticity was enabled

---

## Author Comment (AC2) · 30 Jul 2016

*# General comments The paper presents several innovations for the implementation of coupled near-wake/far-wake model. The model is described in details and compared to a free wake vortex code. The method appears to improve the BEM model at a low computational cost.*
*The new model accounts for the difference in time scales of the far wake model depending on the operating conditions and introduce advanced unsteady aerodynamics corrections. The authors address convergence issues and provide an analytical expression for the determination of the relaxation factor which is essential to reach convergence. Computational time is improved by using this "optimal" relaxation factor and by reducing the number of exponential components of the indicial function. The innovations presented in the paper appear to improve previous implementations. The comparison to a free wake vortex code is relevant and the investigations related to the aerodynamic damping is of great importance for turbine fatigue loads.*
*As a general comment, the inherent differences between the two models should be addressed (either in a discussion paragraph, or by adding extra-investigations), in particular: lifting-line versus lifting-surface approach, rigid curved filaments with no expansion versus free particle, differences in dynamic stall implementations. Each of these "model" differences will have an impact on the results, but the extent of this impact is uncertain. The paper could have even greater value if the impact of this modeling choices were investigated: using a lifting formulation in GENUVP, by deactivating the dynamic stall in both models, etc.*

- We propose to insert the following text before Section 8.2.1:

**Section 8.2.0: Inherent differences between coupled near and far wake model and free wake code**

This paragraph contains an overview of the inherent modeling differences between the different models. It is also detailed how these differences are investigated in the following comparisons of aerodynamic response to pitch steps and prescribed blade vibrations.

The free wake code uses a lifting surface approach, while the near wake model uses a lifting line approach. The free wake code models the airfoils as the camber line of the airfoil sections (thin airfoil approach) using potential theory. Thereby the different sections of the blade have a lift gradient of $2\pi$ and zero drag. Drag and the lift gradient deviation from $2\pi$ is then added afterwards as part of the ONERA unsteady airfoil aerodynamics model. The faster models, on the other hand, all use the airfoil data directly. Thus the measured lift gradients are used for both the shed vorticity model and the near wake model. The influence of this difference is evaluated in the vibration comparisons section by setting the lift gradient to $2\pi$ and drag to zero in case of an edgewise vibration, where drag contributes significantly to the aerodynamic work.

The time constants for the shed vorticity model in the BEM-based codes do not contain a correction for airfoil thickness, but are instead the approximations for a flat plate originally obtained by Jones. The flat plate approximation agrees with the thin lifting surface in the free wake simulations. The comparisons of the aerodynamic response to prescribed blade vibrations contain also results from a BEM model with deactivated shed vorticity model. These results are included to evaluate the isolated influence of the

shed vorticity modeling and distinguish the dynamic effects of shed and trailed vorticity. In the free wake code, the shed vorticity is inherently modeled and can not be turned off.

In contrast to the fast dynamic effects due to trailed and shed vorticity close to the blade, there are also slow dynamic inflow effects. This term is used here to describe effects that would also be visible in actuator disc simulations where the individual blades are not modeled at all. The slow dynamic effects are modeled directly in the free wake code and by means of a dynamic inflow model in the other codes. The influence of these effects is compared for pitch steps where the free wake code results directly model the influence of wake expansion. In case of the blade vibrations, on the other hand, the main effects occur in the direct wake close to the blade. The influence of dynamic inflow in these cases is very small, which is reflected in the large time constants in the modeling. Wake expansion is also expected to be of minor importance in these cases.

Another difference between the free wake code and the BEM-based models is that the dynamic interaction between a blade and the wake of the other blades is only modeled in the free wake code. This introduces dynamic variations in the pitch step cases that are missing in the BEM based codes. If this blade-wake interaction would play an important role in the vibration cases, the agreement between the codes should be better in the high wind speed cases with a larger helix angle than in the low wind cases, where the wake of previous blades is closer to the rotor plane when a blade is passing.

In order to avoid additional uncertainties due to dynamic stall modeling, all cases have been chosen such that stall is mostly avoided. To obtain this, the pitch steps are conducted from 5 degrees to feather towards normal operation, and the amplitudes in the vibration cases are generally small. There may be stall at the very root in the prescribed vibration cases, but the amplitude there is almost zero. Thus there is very little contribution from the root section to the aerodynamic work and differences in dynamic stall modeling do not visibly change the results.

*Other comments are listed below.*
*# Specific Comments p2 l8-9 and p2 l22-26: It seems you are giving results in these sentences. If they are results from this paper, you could consider moving them outside of the introduction part, or remove them completely if they are mentioned further in the document.*

- We think it is acceptable to mention a few key findings at the end of the introduction section.

*p4 l18: Could you elaborate more on the correction for the helix angle? The correction is not part of 3a 3b, is it? The sentence seems to say the correction is applied "on" the helix angle. Does the corrected model still assumes that the vorticity lays in the rotor plane? These were questions that came to me when reading this sentence. They are answered in the given reference but some details could help the reader at that point.*

*(In fact you address this later on, in section 3. But the reader can be confused when it is introduced in section 2).*

- We will add a sentence explaining this briefly in Section 2.

*p4 equation (4): This is understood as a sum of vectors. In this paper, a visual distinctions between vectors and scalars could help the reader.*

- We will add that distinction in the revised article.

*p4 l20 and l24-25: a reference is needed for the BEM model implementation and the a-CT relation used.*

- We will add a reference to "Madsen HA, Bak C, Døssing M, Mikkelsen R, Øye S. Validation and modification of the Blade Element Momentum theory based on comparisons with actuator disc simulations. *Wind Energy* 2010; **13**: 373–389."

*p4 l25-28: The description was hard to follow for me. When it is written "to account for the near wake induction" is it meant in the sense "to compute a 'reduced far-wake' induction which does not include the effect of the near wake induction"? My own reformulation is poor, but could you reformulate the lines 25-28 to guide the reader? Also, could you add a reference or justification to support the removal of the tip-loss factor? Further, though understood, it is probably partially true that a_FW was computed without tip-loss corrections, since the thrust coefficient was probably computed using loads on the blades which in turn were determined included a tip-loss factor. If this is correct, could you comment on that, maybe add an equation to clarify what is meant here?*

- Your reformulation is correct. The removal of the tip loss factor is justified in [Pirrung, G. R., Madsen, H. A., Kim, T., and Heinz, J.: A coupled near and far wake model for wind turbine aerodynamics, Wind Energy, doi:10.1002/we.1969, 2016.]. In the HAWC2 BEM formulation (without near wake model) C_T is computed based on the blade loads, and then modified by applying a tip loss factor. The tip loss factor increases C_T towards the tip, which results in a higher induction towards the tip (due to the a-C_T relation). In the coupled model, this higher induced velocity towards the tip is included in the near wake model induction, and a tip loss correction on the far wake induction factor is not necessary. The thrust coefficient follows from the blade loads, which follow from the velocity triangle, where the sum of near wake and far wake induction is used.

*p9 l9: The author uses a cosine distribution which has indeed some numerical advantages for "elliptic" wings. Number of sections up to 100 were used in this study. Is the model stable above this number of sections? Has the author tried 200 or more sections? Could the elliptic distribution introduce convergence problems or numerical issues due to the small radial elements at the extremities? Have you investigated linear distributions and observed loss of performances (more iterations/computational time, decreased accuracy at the extremities, problems of convergence)? Could the linear distribution then out-perform the elliptic distribution for large number of sections?*

*These questions are slightly rhetorical but your experience on the topic could be beneficial for the community.*

- Larger amounts of sections have been tried:
    - As you expect, there will be convergence problems due to the then very small size of the elements at root and tip. These could be removed by introducing a vortex core radius, which is otherwise not necessary because the prescribed wake prevents vortex filaments from moving extremely close to each other.
    - The computational cost increases roughly with the square of the number of sections, so the near wake computations on 200 sections would be roughly 44 times as expensive as the computation on 30 sections. If for 30 sections the near wake model makes aeroelastic computations roughly 10% slower in normal operation, for 200 sections it would make the computation roughly 440 % slower (so the computation would take more than 5 times longer). Also the convergence is expected to be slower due to the tight spacing, even if a vortex core is included. Thus the additional accuracy by the large number of sections is most likely not enough to justify the large increase in computational effort. We would say that more than the 120 sections used in the section describing the convergence issues are not necessary or advisable for aeroelastic computations.
- The comparison of different point distributions for elliptic wings has been shown in [Pirrung, G. R., Hansen, M. H., and Madsen, H. A.: in: Improvement of a near wake model for trailing vorticity, Proceedings of the science of making torque from wind, Oldenburg, 2012]. There, the distributions are called 'equidistant (which we assume is the linear distribution you refer to), 'cosine' (with sections in the middle of cosine-distributed vortex trailing points) and 'full cosine' (with sections and vortex trailing points distributed at equi angles). In the present paper, the 'full cosine' distribution is used. In the previous torque article it is shown that the induced velocities at the extremities are not computed accurately by a linear distribution, no matter how many sections are used. For recent computations on a blade equipped with flaps, we used a higher resolution, equidistant spacing in the vicinity of the flap and a full cosine distribution on the rest of the wing. The change of distribution is happening where no strong vorticity is trailed, and no discontinuity is visible in the results. In this case, we obtained a very good resolution of the induction dynamics around the flap with a total of 90 aerodynamic sections on the blade. Just increasing the number of sections of a cosine distribution would have led to many more total sections on the wing to obtain good resolution in the region around the flap. So there are cases where we found an equidistant distribution preferable over a full cosine distribution at least on parts of the blade.

*p14 l11: It is not obvious what the "hybrid wake" approach is referring to. Does it refer to the way the far wake influence is computed? Or the combination of dipoles and particles?*

- We will add a sentence: 'Hybrid wake refers to the mixed formulation used in the representation of the wake. In this formulation, the dipole representation is retained for the near part (equivalent to horseshoe filaments) while the far part is modeled by free vortex particles.'

*p15 figure11: The figure demonstrates the effect of the iterative procedure. It could add value to also compare the results with other studies of the acceleration of a flat plate from rest. It can give insights into the validity of the physics and time constants of this model (which was mainly tuned for rotors).*

- For large radii and close spacing (as in the case shown in figure 8) the time constants are approaching those for a straight wing. So the influence of the 'rotor specificness' of the model in this case is negligible (see for example previous examinations for an elliptical wing in [Pirrung, G. R., Hansen, M. H., and Madsen, H. A.: in: Improvement of a near wake model for trailing vorticity, Proceedings of the science of making torque from wind, Oldenburg, 2012])

- The 2D time constants are either according to Jones [Jones, R. T., "The unsteady lift of a wing of finite aspect ratio," Tech. Rep. 681, National Advisory Committee for Aeronautics (United States Advisory Committee for Aeronautics), 1940.] for the lift and drag and according to the panel code used in [Bergami, L., Gaunaa, M. and Heinz, J. (2013), Indicial lift response function: an empirical relation for finite-thickness airfoils, and effects on aeroelastic simulations. Wind Energ., 16: 681–693. doi:10.1002/we.1516] for the unsteady circulation. These 2D constants have been validated previously and their validation lies outside the scope of the present (already large) article.

- A comparison of the dynamic behavior of the near wake model (extended to handle the case of a parked blade) with steady and unsteady NREL/NASA Ames Phase VI measurements has recently been finished and will hopefully soon be published.

*p17+: Oscillations are observed in the results of the GENUVP code. It could add value to the analysis to consider a case where the dynamic stall model was removed from both codes in order to compare the build-up of inductions from both codes. The empiricism from the dynamic stall models introduces an extra level of complexity and thus more uncertainty on the source of the observed differences.*

- The flow is attached in all cases in the paper (except from the very root section, where the oscillations have almost zero amplitude and the pitch response is not investigated). To ensure attached flow and remove dynamic stall related uncertainty is why the pitch steps have been conducted from 5 degrees towards feather to 0 degrees pitch. There is actually no dynamic stall model used in the coupled model computations in the present article, but shed vorticity effects have to be modeled in a 2D unsteady aerodynamics model. This is because the GENUVP code inherently includes shed vorticity, so to be able to compare with GENUVP, shed vorticity needs to be taken into account in the coupled model.
- The oscillations are due to the individual blades passing by the wake of the previous blades in GENUVP, which is not modeled by the other codes.
- This is clarified in the additional section '8.2.0' we suggested to add to the article.

*p19 l15: How are the vibrations prescribed for the GENUVP code?*

- Not only deflection velocities but also deformation of the blade shape is considered. However since deflection is very small (at most 1.5% of the blade radius) we do not expect any secondary effects.

*# Technical corrections*
*p1 l20: I personally had trouble reading the second part of the sentence. Maybe you could replace "far wake computation is using", by "the far wake contribution is computed using"?*

- We will do that.

*p2 l17: Consider splitting the sentence which is a bit long.*

- We will do that.

*p14 l13-14: The sentence "If the relaxation factor..." seems incomplete.*

- This is true, a 'then' will complete the sentence.

*p14 l16: The term "free flow velocity" may be reformulated to "local velocity" to avoid confusion with the free stream velocity.*

- We will change it to 'local flow velocity'.

*p15 l1-3: The sentence is difficult to read, could it be reformulated? Also, consider replacing "local flow angle of attack" to "angle of attack" (or "3D angle of attack" or "local angle of attack").*

- We can change it to: 'Then, viscous corrections are applied to the potential sectional loads that require as input the local flow velocity and angle of attack at every section. In particular, ONERA …'

*p15 l9-11: Reformulation of these lines could help the reading.*

- We propose to write: 'In case of a flexible blade, flow equations are solved for the deformed blade geometry while deformation velocities are accounted for in formulating the non penetration boundary condition.'

*p23 l21: Maybe "and zero drag" is meant. The structure of the sentence is hard to follow.*

- We will replace it by 'and non-zero drag'